# Effects of iron supplements and iron-containing micronutrient powders on the gut microbiome in Bangladeshi infants: a randomized controlled trial

Andrew Baldi ⬤[1,2] ✉, Sabine Braat ⬤[1,3,4], Mohammed Imrul Hasan[1,2,5], Cavan Bennett ⬤[1,2], Marilou Barrios ⬤[6], Naomi Jones[1], Imadh Abdul Azeez ⬤[1,2], Stephen Wilcox[6], Pradip Kumar Roy[1,7], Mohammad Saiful Alam Bhuiyan[5], Ricardo Ataide[1,4], Danielle Clucas[1,2,8], Leila M. Larson ⬤[9], Jena Hamadani[5], Michael Zimmermann ⬤[10], Rory Bowden ⬤[1,2,6], Aaron Jex[1,2,7], Beverley-Ann Biggs[4,11] & Sant-Rayn Pasricha ⬤[1,2,8,12] ✉

Anemia is highly prevalent globally, especially in young children in low-income countries, where it often overlaps with a high burden of diarrheal disease. Distribution of iron interventions (as supplements or iron-containing multiple micronutrient powders, MNPs) is a key anemia reduction strategy. Small studies in Africa indicate iron may reprofile the gut microbiome towards pathogenic species. We seek to evaluate the safety of iron and MNPs based on their effects on diversity, composition, and function of the gut microbiome in children in rural Bangladesh as part of a large placebo-controlled randomized controlled trial of iron or MNPs given for 3 months (ACTRN12617000660381). In 923 infants, we evaluate the microbiome before, immediately following, and nine months after interventions, using 16S rRNA gene sequencing and shotgun metagenomics in a subset. We identify no increase in diarrhea with either treatment. In our primary analysis, neither iron nor MNPs alter gut microbiome diversity or composition. However, when not adjusting for multiple comparisons, compared to placebo, children receiving iron and MNPs exhibit reductions in commensal species (e.g., *Bifidobacterium, Lactobacillus*) and increases in potential pathogens, including *Clostridium*. These increases are most evident in children with baseline iron repletion and are further supported by trend-based statistical analyses.

Anemia affects almost 300 million pre-school-aged children worldwide, and is most prevalent in children in low- and middle-income countries (LMICs).[1] In 2019, the prevalence of anemia in children aged 6–59 months was 49.0% in South-East Asia, and 43.1% in Bangladesh.[2] Anemia in young children is often accompanied by a concomitant burden of diarrhea in low-income settings. Diarrhea

causes 10% of all child deaths in Asia and Africa,[3] precipitated by unsafe water, sanitation and hygiene (WASH) conditions. The World Health Organization (WHO) recommends universal distribution of iron (either as iron-containing multiple micronutrient powders– MNPs, or iron supplements, e.g., drops) to all young children where anemia is prevalent.[4,5] However, the safety of this approach is

uncertain, with large field trials of iron-containing MNPs suggesting an increased risk of diarrhea.[6]

Trials in sub-Saharan Africa have reported pathogenic reprofiling of the gut microbiome (an increase in pathogenic enterobacteria and a decrease in commensal species) in young children receiving iron-containing MNPs, associated with evidence of increased intestinal inflammation. This may relate to an increase in colonic iron that facilitates the growth of potentially pathogenic bacteria at the expense of commensal species, leading to dysbiosis.[7] An effect of iron interventions on the microbiome has been observed in studies conducted in sub-Saharan Africa. A study of Kenyan children receiving iron-containing MNPs reported that iron promoted growth of enteropathogens e.g., *Escherichia*, *Salmonella* and *Clostridium*, and increased intestinal inflammation.[8] A second Kenyan study reported that even low-dose iron pathogenically reprofiled the microbiota and exacerbated intestinal inflammation.[9] In both cases, these conclusions were drawn from statistical analyses of the microbiome that were not adjusted for multiple comparisons, and could represent false discovery (i.e., false positive) findings. Crucially, there remains limited data evaluating microbiome effects in South Asia, where effects from iron on the clinical endpoint of diarrhea have been observed.[6] Iron status (deficiency or repletion) influences intestinal iron uptake[10] and could, therefore, potentially modify the effects of interventions on the microbiome.

Defining the safety of iron interventions is critical, given the ongoing public health recommendation for their distribution.[11] There is an urgent need for well-powered, placebo-controlled clinical trials that evaluate the safety of iron interventions on the gut microbiome using a rigorous study design to examine causal relationships.

To address this, we leverage the BRISC (Benefits and Risks of Iron InterventionS in Children) trial, a large, placebo-controlled double-blind, double-dummy randomized controlled trial conducted in rural Bangladesh in which infants were randomized to receive three months of daily iron syrup, MNPs or placebo.[12] In this large sub-study we present here, we evaluate stool samples from a subset of BRISC participants at baseline, after three months of intervention, and after a further 9-month post-intervention follow-up. Samples are analyzed using 16S rRNA amplicon sequencing, with a subset also analyzed with shotgun metagenomic sequencing. This combined methodology enables us to define causal iron-induced gut microbiome reprofiling by exploring the effects of iron and MNPs compared to placebo on diversity and composition across an unprecedented sample size while also undertaking a high-resolution evaluation of the effect of iron on species and function.

## Results

Between September 2018 and February 2019, we collected 923 baseline stool samples, 796 samples at the subsequent post-intervention time point 13 weeks later, and 578 samples at the post-follow-up time point (Fig. 1A, Supplementary Fig. 1). Overall, a sample was provided at baseline from 84% of participants approached for this sub-study, and of these, 86% provided a sample at midline and 63% at endline. Baseline characteristics of the cohort are summarized in Table 1. 16S rRNA amplicon sequencing was performed on 923 samples at baseline, 796 at midline, and 578 at endline; a subset of 319, 320 and 315 samples also underwent shotgun metagenomic sequencing at these time points, respectively (Fig. 1B). Baseline characteristics of the shotgun metagenomic sub-cohort are summarized in Table S1 (*Supplementary Material*).

The BRISC trial found that across all participants, neither iron nor MNPs significantly increased parent-reported days with diarrhea (incidence rate ratio (IRR) 1.13 [95% CI 0.89–1.42] for iron versus placebo, IRR 1.17 [0.93–1.48] for MNPs versus placebo).[12] Among children in the microbiome sub-study, again, neither iron nor MNPs statistically significantly increased parent-reported days with diarrhea (IRR 1.43

[0.89–2.27] for iron versus placebo, IRR 1.24 [0.77–1.98] for MNPs versus placebo).

For the gut microbiome analysis we first examined baseline taxonomic profiles and changes with age. The top five genera detected at baseline are presented in Fig. 1C (source data in *Supplementary Material*). No significant baseline differences were seen between arms at the genus or species levels. We then examined changes to the microbiome as children grew from 8 to 20 months of age. Using 16S and shotgun metagenomics, we observed reductions in Actinomycetota and Pseudomonadota phyla and increased Bacteroidota over this 12-month timeframe (Fig. 1D, E). Compared to baseline, many key functional pathways altered as children reached the 3-month post-intervention period (11 months of age); 155 pathways were differentially abundant compared to baseline (91 increased and 64 reduced). At 9 months follow-up (age 20 months), 349 pathways were differentially abundant (113 up and 236 down) compared to baseline. For both comparisons the pathway with the greatest increase was dTDP-L-rhamnose biosynthesis I ($\log_2$-fold change 0.001, SD 0.0004, adj. $p$-value 0.003 at 11 months; $\log_2$-fold change 0.006, SD 0.0004, adj. $p < 0.001$ at 20 months) and greatest reduction was glucose and glucose-1-phosphate degradation ($\log_2$-fold change −0.002, SD 0.0003, adj. $p < 0.001$ at 11 months; $\log_2$-fold change −0.005, SD 0.0003, adj. $p < 0.001$ at 20 months) (Fig. 1F, with additional pathways detailed in Supplementary Fig. 2 and in source data in *Supplementary Material*).

Next, using 16S rRNA gene sequencing data, we evaluated the effects of iron interventions on microbial diversity. Neither iron supplements nor MNPs altered alpha diversity (Shannon and inverse Simpson indices) at the immediate post-intervention time point. Likewise, we did not find an effect from the interventions on beta diversity (Fig. 2A, B). Furthermore, there were no differences between trial arms in abundance at the genus level, after FDR adjustment. Shotgun metagenomic analysis by the trial arm revealed no significant changes in differential abundance at the species level, nor by functional pathways measured by *HUMAnN 3.0* (Fig. 3A, B), after FDR adjustment (source data in *Supplementary Material*).

We reasoned that iron interventions may differentially impact the gut microbiome in iron-deficient compared with iron-replete children, as intestinal iron absorption may be higher in children with iron depletion (mediated by suppressed hepcidin levels),[13,14] leaving less luminal iron to influence the microbiome.[15] To evaluate this, we undertook a subgroup analysis by baseline iron status on the effect of oral iron interventions on the microbiome. We defined iron deficiency in two ways: (1) evidence of low baseline iron stores (iron deficiency, defined as a ferritin concentration <12 μg/L or <30 μg/L if C-reactive protein >5 mg/L, accounting for effect of inflammation on ferritin)[12,16] compared to iron repletion, and (2) evidence of increased iron uptake by suppressed baseline hepcidin (<10 ng/mL) compared to non-low hepcidin (10 ng/mL or higher).[17] Using 16S rRNA gene sequencing data, no differences in alpha diversity were observed between iron, MNP and placebo groups comparing baseline iron-deficient and iron-replete subgroups or baseline low hepcidin and non-low hepcidin (Fig. 4A–D). Likewise, we did not detect any significant differences at genus or species levels between trial arms in any subgroup, nor did we detect a difference in effect from iron between these subgroups in functional analysis using the shotgun metagenomic data, all using FDR-adjusted analyses.

Previous studies reporting effects of iron on gut microbiome profile have evaluated differential abundance of particular taxa without adjusting statistical significance thresholds for false discovery.[8,9] Importantly, our study sought to evaluate the safety of a public health intervention that is distributed to children worldwide. We therefore also report findings using unadjusted $p$ values to evaluate any differential abundance in gut genera or species between trial arms. Through this analysis, by 16S rRNA gene

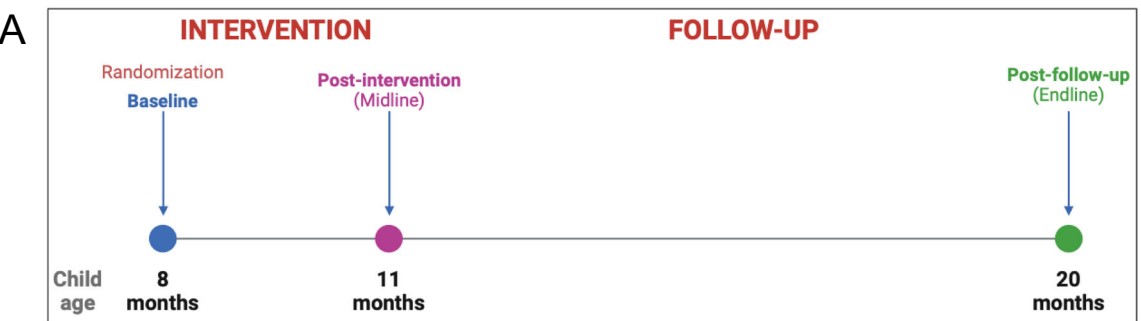

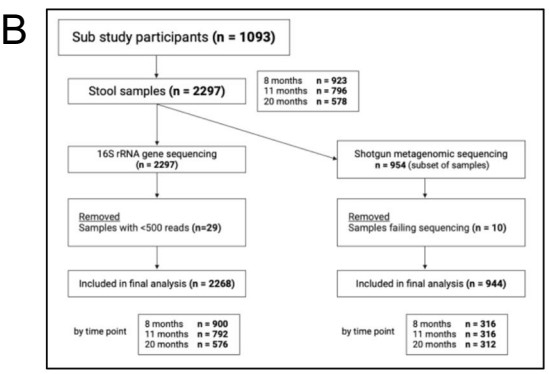

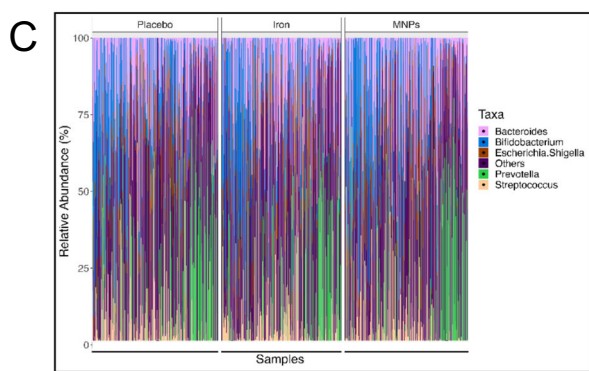

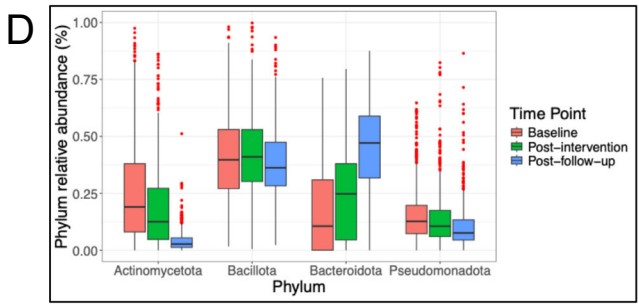

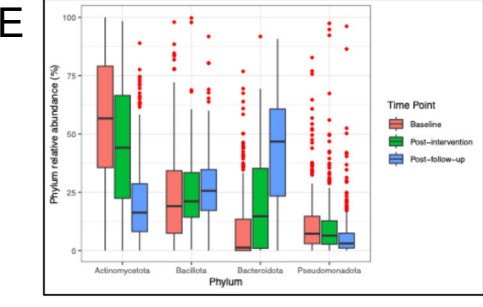

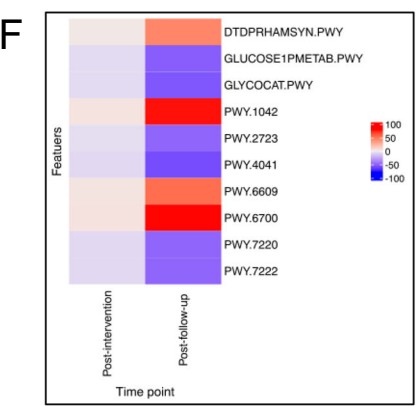

sequencing, no genera were found to be differentially abundant between the intervention arms compared to placebo at the midline time point (Fig. 5A). However, shotgun metagenomic unadjusted analysis at the species level indicated that iron supplementation was associated with higher abundance of *Enterococcus faecalis* (log$_2$-fold change 0.12, SD 0.04, $p = 0.002$, FDR-adjusted $p = 0.452$) at the midline time point, while children who received MNPs had small

increases in *Clostridium* species (*C. saccharolyticum* log$_2$-fold change 0.08, SD 0.04, $p = 0.026$, FDR-adjusted $p = 0.994$ and *C. neonatale* log$_2$-fold change 0.04, SD 0.02, $p = 0.048$, FDR-adjusted $p = 0.994$) and reductions in *Lactobacillus gasseri* (log$_2$-fold change −0.07, SD 0.04, $p = 0.049$, FDR-adjusted $p = 0.994$) and *Escherichia coli* (log$_2$-fold change −0.27, SD 0.13, $p = 0.040$, FDR-adjusted $p = 0.994$) (Fig. 5B).

**Fig. 1 | Study design, sample information and analysis by time point. A** BRISC trial schema showing assessment and sampling time points. **B** Flow diagram outlining stool samples for 16S rRNA and shotgun metagenomic sequencing. **C** Stacked bar plot presenting taxonomic composition of baseline samples at study entry (i.e., baseline) based on relative abundance of the top five genera overall. Each bar represents a sample, with relative abundance on the y-axis. Samples are grouped by trial arm (16S rRNA data). Taxonomic boxplots showing relative abundance of the four most abundant bacterial phyla by sampling time point (**D**) 16S rRNA data and (**E**). shotgun metagenomic data. Boxplot center lines denote median value, with bounds of box indicating 25–75th percentiles. Whisker lines encompass 1.5 x interquartile range from above the upper and below the lower quartiles. Data points outside whiskers are outliers. (*n* = 900 at baseline, *n* = 790 at post-intervention and *n* = 574 at post-follow-up time points for (**D**), and *n* = 316 at baseline, *n* = 316 at post-intervention and *n* = 312 at post-follow-up time points for

(**E**). **F** Heatmap showing relative differences in abundance of genes relating to microbiome functional profile by sampling time point across all trial arms, with baseline as reference. Relative abundance data underwent centered log ratio normalization/transformation in a general linear model analysis in MaAsLin2, with data for differentially abundant functional pathways including coefficient (approximating log2-fold change), standard deviation and FDR-adjusted *p* value. Only the 10 features exhibiting the greatest relative change in abundance (increase – in red – or decrease – in blue – from baseline) are shown. Abundance is expressed as $-\log_{10}$(FDR-adjusted *p* value) * sign(log$_2$-fold change) as per the heatmap calculation used by *MaAsLin2*. (*n* = 255 at baseline, *n* = 231 at post-intervention and *n* = 312 at post-follow-up time points) (Shotgun metagenomic data). **A**, **B** created with BioRender.com released under a Creative Commons Attribution-NonCommercial-NoDerivs 4.0 International license (https://creativecommons.org/licenses/by-nc-nd/4.0/deed.en). Source data for Fig. 1C–F are provided in the Source Data file.

## Table. 1 | Baseline characteristics of participants in the BRISC microbiome sub-study

| | Iron<br>*N* = 308 | MNPs<br>*N* = 307 | Placebo<br>*N* = 308 |
|---|---|---|---|
| Union | | | |
| Bhulta | 98/308 (31.8%) | 100/307 (32.6%) | 100/308 (32.5%) |
| Golakandail | 112/308 (36.4%) | 112/307 (36.5%) | 106/308 (34.4%) |
| Rupganj | 98/308 (31.8%) | 95/307 (30.9%) | 102/308 (33.1%) |
| Household with food-secure status[a] | 242/307 (78.8%) | 245/304 (80.6%) | 251/305 (82.3%) |
| Age extra food in addition to breastfed (months), median (IQR) | 6.0 (5.0–6.0) | 6.0 (5.0–6.0) | 6.0 (5.0–6.0) |
| Hemoglobin concentration (g/L) venous, mean (SD) | 110.2 (10.3) | 110.2 (9.6) | 109.8 (9.2) |
| Anemia venous[b] | 140/303 (46.2%) | 127/298 (42.6%) | 135/299 (45.2%) |
| Ferritin (ug/L), median (IQR) | 22.6 (11.7–35.1) | 23.6 (13.1–37.2) | 23.6 (13.2–38.4) |
| Iron deficient[c] | 95/294 (32.3%) | 74/288 (25.7%) | 78/290 (26.9%) |
| Iron deficient anemia venous[d] | 69/294 (23.5%) | 46/288 (16.0%) | 60/290 (20.7%) |
| Hepcidin – (ng/mL), median (IQR) | 27.2 (16.0–46.2) | 28.7 (14.2–55.5) | 33.2 (17.2–53.8) |
| Low hepcidin (hepcidin <10 ng/mL) | 27/227 (11.9%) | 34/229 (14.8%) | 22/229 (9.6%) |
| C-reactive protein (mg/L), median (IQR) | 0.82 (0.32–2.65) | 0.78 (0.31–2.34) | 0.81 (0.32–2.85) |
| Inflammation[e] | 43/294 (14.6%) | 36/288 (12.5%) | 44/290 (15.2%) |
| Length/height-for-age z-score, mean (SD) | –1.30 (1.00) | –1.26 (1.05) | –1.35 (0.96) |
| Stunted[f] | 75/308 (24.4%) | 77/305 (25.2%) | 65/308 (21.1%) |
| Weight-for-age z-score, mean (SD) | –0.53 (1.04) | –0.53 (1.09) | –0.57 (0.94) |
| Underweight[g] | 21/308 (6.8%) | 25/305 (8.2%) | 22/308 (7.1%) |
| Weight-for-length/height z-score, mean (SD) | 0.36 (1.00) | 0.32 (1.00) | 0.34 (1.00) |
| Wasting[h] | 2/308 (0.6%) | 3/305 (1.0%) | 2/308 (0.6%) |

Data are n/N (%) unless stated otherwise.
*MNPs* Micronutrient powders, *SD* Standard Deviation, *IQR* Interquartile range (25th to 75th percentile).
[a]Household food security was assessed and defined using the Household Food Insecurity Access Scale.
[b]Anemia was defined as venous hemoglobin <110 g/L.[1]
[c]Iron deficiency was defined as ferritin level <12 μg/L or <30 μg/L if C-reactive protein was (>5 mg/L).[1]
[d]Iron deficiency anemia was defined as concurrent iron deficiency and anemia.
[e]Inflammation was defined as C-reactive protein >5 mg/L.
[f]Stunting was defined as length-for-age z-score <–2.[2]
[g]Underweight was defined as weight-for-age z-score <–2.[2]
[h]Wasting was defined as weight-for-length z-score <–2.[2]

We next report unadjusted analysis to the previously defined baseline iron status subgroups utilizing the shotgun metagenomic data. Children with baseline iron repletion or non-low baseline hepcidin receiving MNPs exhibited an increase in *Clostridium* species after intervention. Iron-replete children receiving iron had a reduction in *Bifidobacterium bifidum* at the same time point (Fig. 5D, F). Likewise, the iron supplementation-related increase in *E. faecalis* was only statistically significant in the non-low hepcidin subgroup (Fig. 5F compared to Fig. 5C, E).

To further interrogate the effects of iron interventions on differential abundance, we finally explored two key phyla previously

reported to change after oral iron interventions.[8] Using shotgun metagenomic data from samples obtained immediately post-intervention, we conducted an enrichment analysis in which species abundance patterns within each of phylum Bacteroidota and phylum Bacillota were compared between iron/MNPs and placebo. We found that Bacteroidota species were significantly enriched with placebo (10 species, net decrease with iron compared to placebo, *p* = 0.01) and Bacillota species were significantly enriched with iron supplements (57 species, net increase to placebo, *p* = 0.04) (Fig. 6A, B). However, no statistically significant effects were observed in the enrichment analysis of MNPs versus placebo for either phylum (Fig. 6C, D).

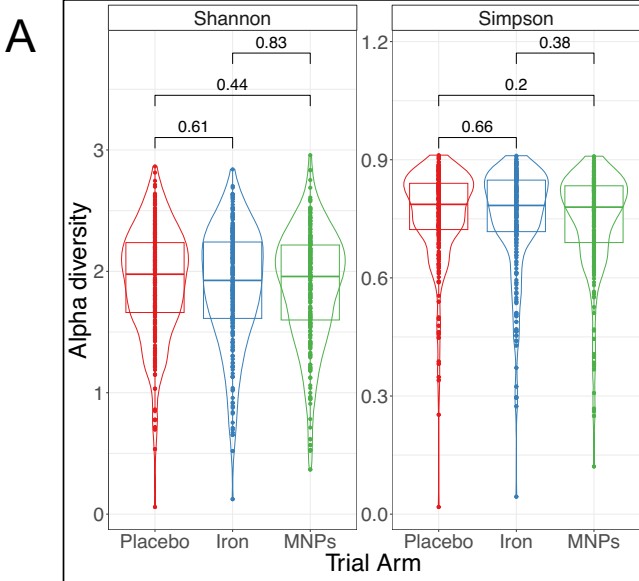

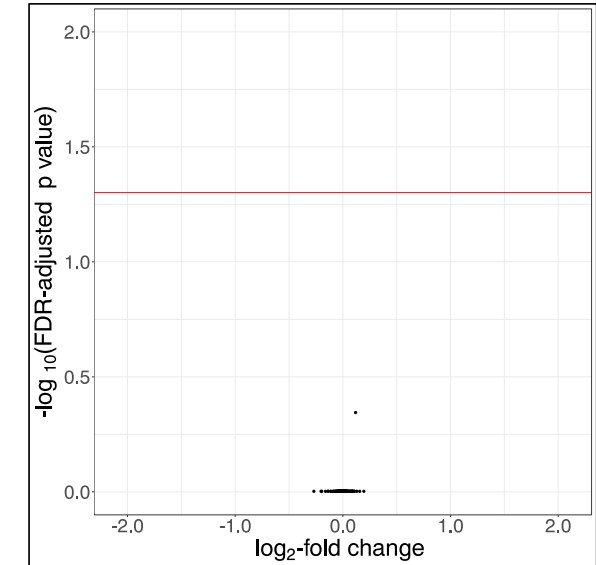

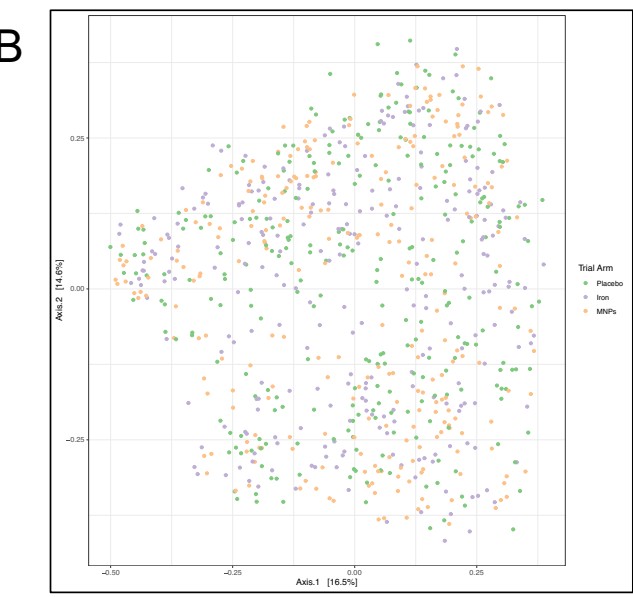

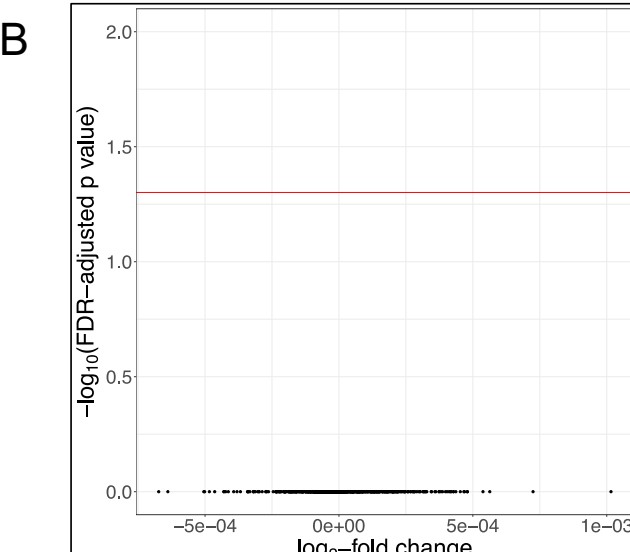

**Fig. 2 | Alpha and beta diversity by trial arm. A** Violin plot presenting taxonomic alpha diversity–measured by Shannon and inverse Simpson indices–by trial arm immediately post-intervention (16S rRNA data). Each dot represents an individual sample, grouped by trial arm. Group differences were calculated using pair-wise ANOVA for each diversity measure. Center lines denote median value, with rectangles showing 25–75th percentiles. The violin outlines the distribution of the data, with wider sections representing a higher probability that samples in the dataset will have the corresponding value and narrower sections representing a lower probability. **B** Principal coordinates analysis plot illustrating beta diversity of samples by trial arm post-intervention and measured by Bray-Curtis dissimilarity, with permutational analysis of variance (PERMANOVA) used as the statistical test ($p = 0.841$) (16S rRNA data). Each dot represents an individual sample, grouped by trial arm. ($n = 276$ for placebo, $n = 254$ for iron and $n = 262$ for MNPs).

**Fig. 3 | Differential abundance diversity and trial arm.** Volcano plots presenting differential abundance by trial arm immediately post-intervention, with (**A**) showing no significant differences at the species level and (**B**) showing no significant differences in abundance of functional pathways measured by *HUMAnN 3.0* (Shotgun metagenomic data). Relative abundance data underwent centered log ratio normalization/transformation in a general linear model analysis in MaAsLin2, with data for differentially abundant species and functional pathways including coefficient (approximating log2-fold change), standard deviation and FDR-adjusted $p$ value. Each figure shows $\log_2$-fold change on the x-axis and the $-\log_{10}$(FDR-adjusted $p$ value) on the y-axis. The horizontal red line indicates an FDR-adjusted $p$ value 0.05. Source data for this figure are provided in the Source Data file.

There was high overall parent-reported adherence to the intervention during the 13-week intervention period, with those with 70% or higher adherence to syrup and sachet 74.7%, 72.3% and 76.9% in the iron, MNPs and placebo arms respectively. A further subset analysis was undertaken to evaluate the effect of adherence on the microbiome. Restricting the analysis to samples from participants with at least 70% adherence to their interventions did not influence results compared to analysis of the overall study population (Supplementary Fig 3A–D).

Finally, we evaluated the effects of interventions 9 months following completion, at the endline time point, using FDR-adjusted analyses and the overall group only. These analyses indicate higher alpha diversity among the MNP group compared to the placebo or the iron group, reduced abundance of *Prevotella copri* in the MNP group compared to placebo, and differential abundance of several genes relating to metabolic pathways (Supplementary Fig 4A–D).

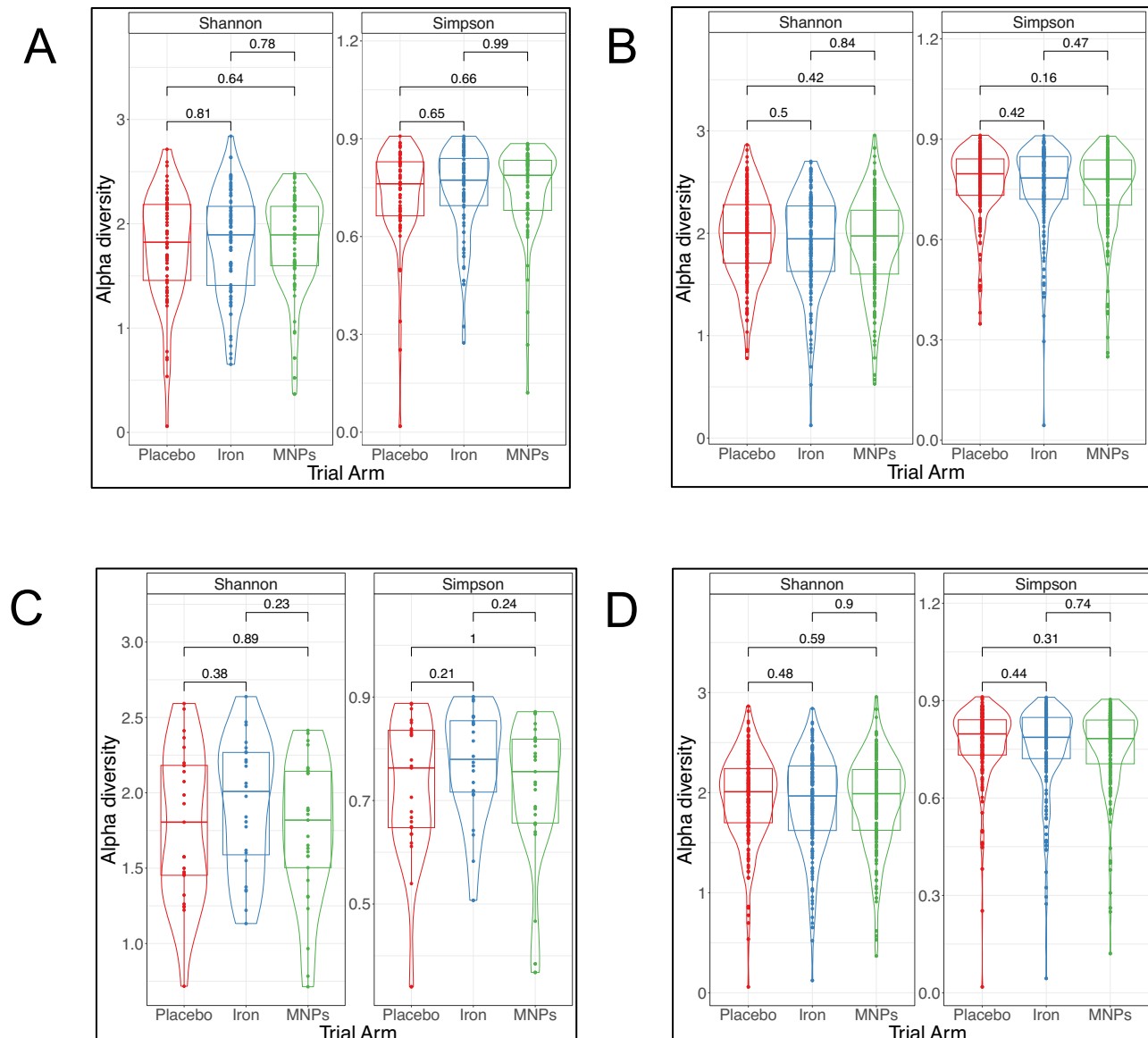

**Fig. 4 | Subgroup analyses of baseline iron status by trial arm.** Violin plots presenting alpha diversity – measured by Shannon and inverse Simpson indices – by trial arm immediately post-intervention according to subgroups: (**A**) participants with baseline iron deficiency, (**B**) participants with baseline iron repletion, (**C**) participants with low hepcidin (<10 ng/mL) at baseline and (**D**) participants with non-low hepcidin (≥10 ng/mL) at baseline. Iron deficiency was defined as either serum ferritin <12 µg/L or <30 µg/L if C-reactive protein was (>5 mg/L). (16S rRNA data). Each dot represents an individual sample, grouped by trial arm. Group differences were calculated using pair-wise ANOVA for each diversity measure. Center lines denote median value, with rectangles showing 25–75th percentiles. The violin outlines the distribution of the data, with wider sections representing a higher probability that samples in the dataset will have the corresponding value and narrower sections representing a lower probability. (n = 67 for placebo, n = 80 for iron and n = 68 for MNPs for (**A**), 195, 170, and 181 for (**B**), 25, 24, and 29 for (**C**), and 205, 189, and 197 for (**D**)).

## Discussion

There is concern regarding the safety and hence net benefit of universal iron interventions for young children living in settings where anemia is highly prevalent but exposure to infectious diseases is intense.[18] To define the effect of iron interventions on the gut microbiome we leveraged a randomized placebo-controlled trial of iron supplements and MNPs in young children in rural Bangladesh, using 16S rRNA gene sequencing across over almost 1000 children, and shotgun metagenomics in a subsample for higher resolution assessment. We found no evidence that iron interventions adversely reprofile the gut microbiome when an unbiased bioinformatic approach with statistical significance adjusted for false discoveries was used. However, unadjusted findings indicate that iron interventions given to children with baseline iron repletion reduce abundance of

commensals and may increase abundance of some potentially pathogenic species. These findings are supported by evidence from shotgun metagenomic data of differential enrichment of two phyla previously linked to iron-microbiome interactions, suggesting these changes may not be due to false discovery errors.

Previous studies in sub-Saharan Africa have shown that iron may increase the prevalence of intestinal pathogens. Children in Côte d'Ivoire randomized to 6 months of iron-fortified biscuits (n = 60) exhibited increased pathogenic enterobacteria (measured using PCR) and intestinal inflammation.[19] In 6-month-old Kenyan children (n = 115) where there was high baseline carriage of pathogenic species (e.g., *C. difficile*, *C. perfringens*, *Salmonella*, pathogenic *E. coli*), MNPs with iron promoted a greater increase (or less of a decrease) in taxa including *Bacillota* (phylum), *Escherichia/Shigella* and *Clostridium* (genera)

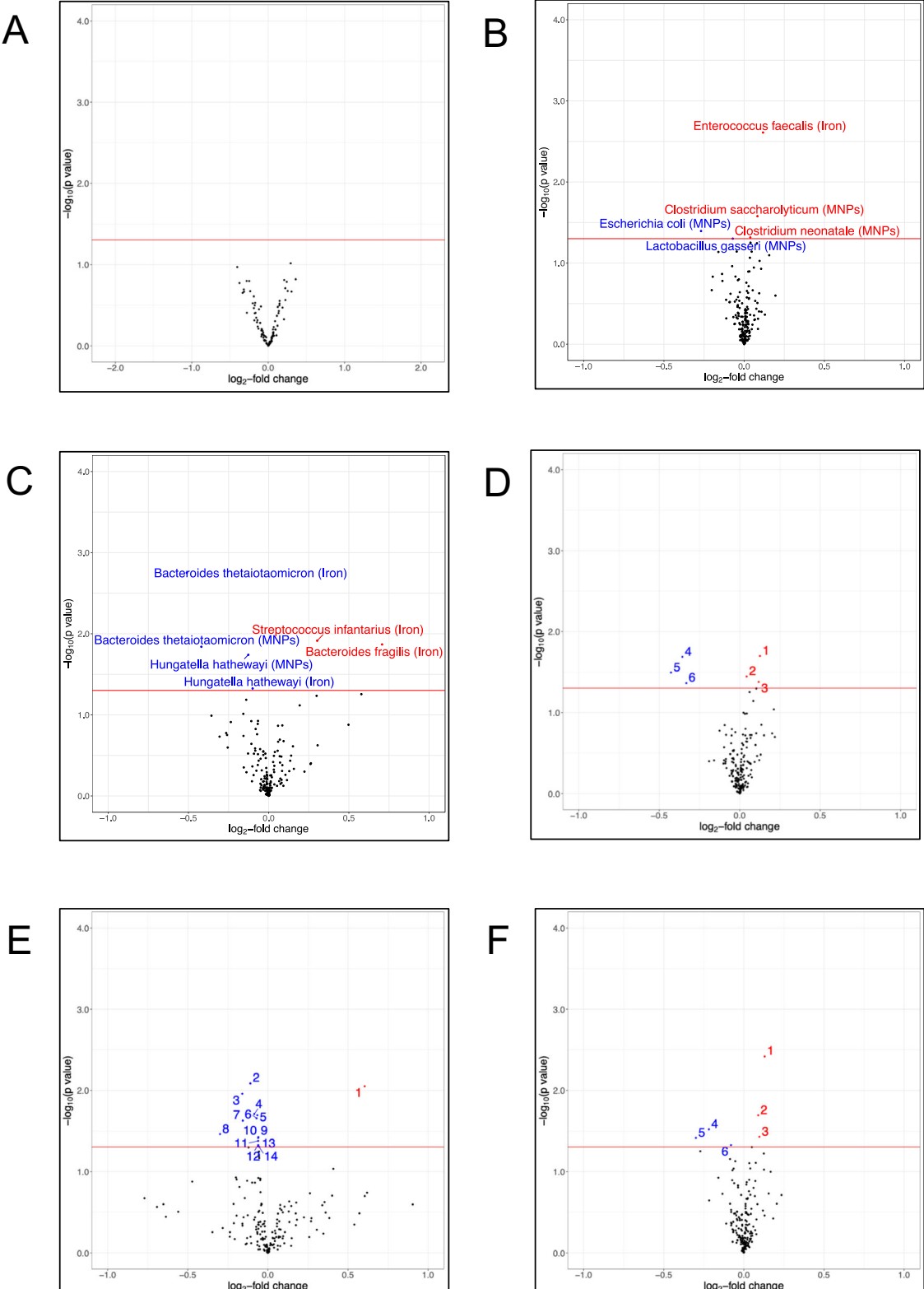

measured using PCR and 16S rRNA gene sequencing.[8] In a third study, also in Kenya ($n = 155$), even low-dose MNPs (5 mg iron) pathogenically reprofiled the microbiota (measured by 16S rRNA gene sequencing and PCR) and exacerbated intestinal inflammation.[9] Notably, differential abundance testing in these studies was not adjusted for false discovery. In contrast to these three trials, several others have not identified an effect of iron on gut microbiome composition.[20–22] For example, iron-fortified lipid nutrient supplements did not reprofile the

gut microbiome in a trial of 213 Malawian infants measured with 16S rRNA gene sequencing.[20] In another small Kenyan study ($n = 33$), *Escherichia* and *Bifidobacterium* abundances in infants who received iron-containing MNPs versus placebo or a non-iron MNP showed no significant differences from control measured using 16S rRNA gene sequencing.[21] Finally, a small South African study ($n = 49$), where baseline prevalence of pathogenic bacteria was low, no adverse effects from iron were observed on microbiota or inflammation by PCR.[22] Our

**Fig. 5 | Unadjusted analysis of differential abundance by trial arm.** Volcano plots illustrating differential abundance by trial arm as measured immediately post-intervention utilizing unadjusted *p* values. **A** Differential abundance at the genus level. **B**. Differential abundance at the species level. Illustrate results of unadjusted differential abundance analyses at the species level of subgroups: (**C**) participants with baseline iron deficiency, (**D**) participants with baseline iron repletion (1. *Clostridium innocuum* (MNPs), 2. *Clostridioides difficile* (MNPs), 3. *C. innocuum* (Iron), 4. *Escherichia coli* (MNPs), 5. *Bifidobacterium bifidum* (Iron), 6. *E. coli* (Iron)) (**E**) participants with low hepcidin (<10 ng/mL) at baseline (1. *Streptococcus infantarius* (Iron), 2. *Lactobacillus vaginalis* (Iron), 3. *Clostridium paraputrificum* (Iron), 4. *Clostridium butyricum* (Iron), 5. *Enterobacter cloacae* complex (Iron), 6. *Clostridium* sp 7 2 43FAA (Iron), 7. *Lactobacillus salivarius* (Iron), 8. *Enterococcus avium* (Iron), 9. *Sutterella parvirubra* (Iron), 10. *Allisonella histaminiformans* (Iron), 11. *Collinsella intestinalis* (Iron), 12. *Actinomyces* sp HPA0247 (Iron), 13. *Bifidobacterium dentium*

(Iron), 14. *Collinsella stercoris* (Iron)) and (**F**) participants with non-low hepcidin (≥10 ng/mL) at baseline (1. *Enterococcus faecalis* (Iron), 2. *Ruminococcus torques* (Iron), 3. *Clostridium innocuum* (MNPs), 4. *Enterococcus faecium* (MNPs), 5. *Escherichia coli* (Iron), 6. *Lactobacillus gasseri* (MNPs)). Iron deficiency was defined as either serum ferritin <12 μg/L or <30 μg/L if C-reactive protein was (>5 mg/L). (**A** 16S rRNA data; **B**–**F**. Shotgun metagenomic data). For (**A**), a log linear regression model is performed on abundance data to determine differential abundance of genera and present unadjusted *p* values. For (**B**–**F**), relative abundance data underwent centered log ratio normalization/transformation in a general linear model analysis in MaAsLin2, with data for differentially abundant species including coefficient (approximating log2-fold change), standard deviation and unadjusted *p* value. Each figure shows $\log_2$-fold change on the x-axis and the $-\log_{10}$(unadjusted *p* value) on the y-axis. The horizontal red line indicates an unadjusted *p* value 0.05. Source data for this figure are provided in the Source Data file.

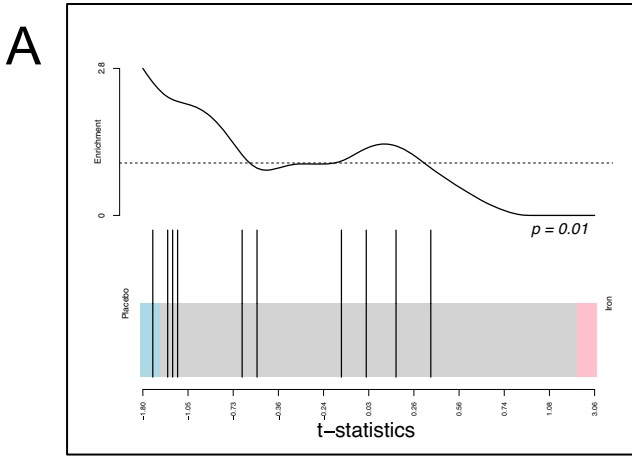

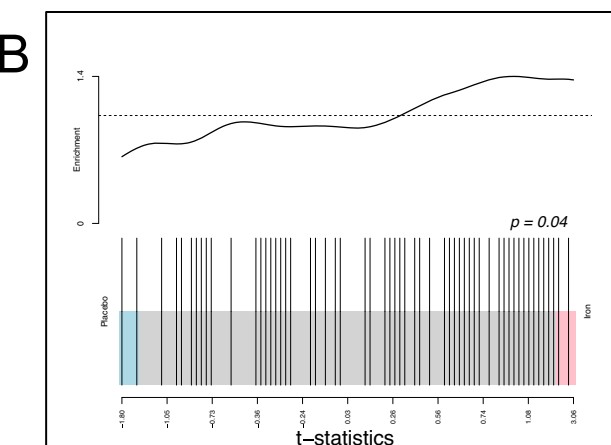

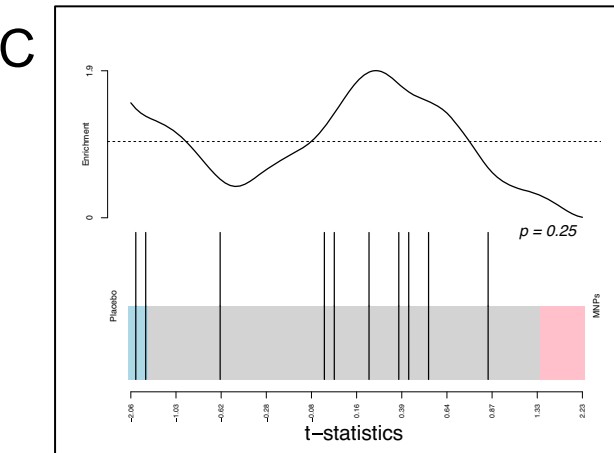

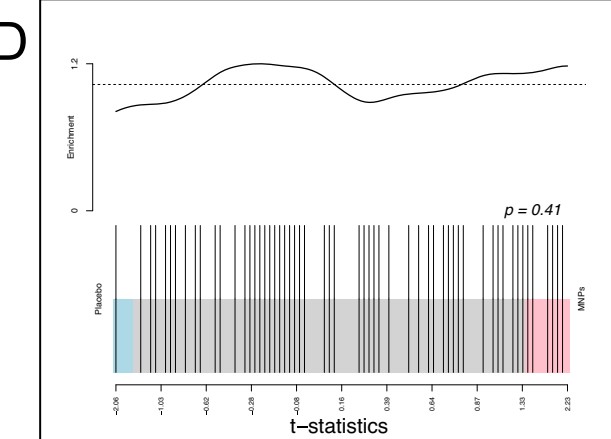

**Fig. 6 | Enrichment analysis.** Competitive phylum enrichment analysis, contrasting the phyla Bacillota and Bacteroidota. Enrichment is assessed using camera PR and t-statistics to rank species based on their differential expression. These are used to determine statistically significant variations in the aggregate mean rank order for the entire phylum between different iron and placebo. In the barcode plots, individual species within a phylum are represented by vertical bars; species

that are up-regulated under placebo conditions appear on the left, while those up-regulated in response to the intervention are on the right. The line above represents the distribution of t-statistics, highlighting enrichment of species within the phyla. **A** Bacteroidota (placebo-iron), (**B**) Bacillota (placebo-iron), (**C**) Bacteroidota (placebo-MNPs) and (**D**) Bacillota (placebo-MNPs) (Shotgun metagenomic data).

16S rRNA gene sequencing study exceeds the cumulative sample size across these studies; in addition, by applying shotgun metagenomics to this problem, our study provides higher sequencing resolution to address this problem.

We recognize the risk of false discoveries when analysing the effects of interventions across hundreds of potential genera or species

and aimed to present conservative findings through the rigorous control of the false discovery rate (FDR) and transparent reporting. Importantly, our analyses represent a safety (rather than efficacy) study, where conservative application of multiple testing corrections may be prudent to avoid the risk of false negative findings.[23] To further challenge our findings, we undertook enrichment analysis that

revealed iron-induced changes within members of Bacillota and Bacteroidota phyla. These tests evaluate overall changes in the distribution of all members of each phylum and do not require correction for multiple testing. Although they do not reveal which species within a phylum are changing with iron treatment, they do show that there is a statistically significant increase in Bacillota and reduction in Bacteroidota with iron relative to placebo.

Our findings exhibit biological plausibility and consistency with previous findings. In the colon, commensal bacteria (e.g., *Lactobacillus* and *Bifidobacterium*) are crucial in preventing colonization by pathogenic species.[24] Lactobacilli do not require iron, and most *Bifidobacterium* species require only little iron compared to other taxa.[8] In contrast, iron is essential for virulence and colonization by enteric pathogens (i.e., gram-negative organisms such as *Salmonella*, pathogenic *E. coli* and *Shigella*). Oral iron supplements (especially iron in MNPs) are only partially absorbed, with most iron reaching the colon, where it can influence the gut microbiome.[7] In our study, we observed possible iron-induced reductions in *Lactobacillus* and *Bifidobacterium* species and increases in *Clostridium* species, broadly consistent with previous positive findings.[8,9] However, in our study, iron did not increase *E. coli* (rather, caused reductions) or other Enterobacteriaceae. The effect size of iron on differential abundance was relatively small, thus, the clinical significance of these changes remains uncertain.

The master iron-regulatory hormone, hepcidin, regulates iron absorption from the intestine.[10] We reasoned that iron-deplete children (with suppressed hepcidin) absorb more iron from the supplement, whereas iron-replete children may absorb less iron, with more iron reaching the colon and influencing the gut microbiome. Overall, iron interventions did not have a differential impact on the gut microbiome between iron-deficient or iron-replete children. However, unadjusted analyses indicated that reprofiling of the microbiome was most profound among children with baseline iron repletion, where higher doses of iron may reach the colon, providing biological plausibility to this finding. These findings were supported by our enrichment analyses. These results may indicate that iron may be best deployed for treatment of iron deficiency or iron deficiency anemia, rather than for universal use. However, a screen-and-treat program would require near-patient testing for iron deficiency prior to provision of iron, which would raise the cost and complexity of this public health program and may leave still some anemia untreated. Such cost-effective testing technology remains an unmet need. This approach was explored in two Gambian randomized trials of pre-iron supplementation hepcidin screening in pregnancy and in infants that did not demonstrate non-inferiority in terms of anemia control.[14,25]

Our study has several strengths. The parent trial was a placebo-controlled (blinded) randomized trial in a rural, low-income South Asian setting, with high adherence. Wet, bioinformatic and statistical analyses were undertaken prior to unblinding of the researchers to the study arms. Crucially, ours is the largest and one the first studies to incorporate measurement of iron biomarkers, including the first to assess the iron-regulatory hormone hepcidin in a subgroup analysis, to enable assessment of the interaction between host iron status (and iron uptake), iron interventions, and effects on the microbiome.

One limitation of our study is that we did not evaluate non-bacterial organisms in the gut microbiome, such as fungi and viruses. Viruses are among the leading causes of diarrheal infections requiring hospitalization in children in Bangladesh, and this is likely the case in many other LMICs.[26] For example, adenovirus is among the most common causes of diarrhea in Bangladesh in both children and adults.[26] Future analysis will evaluate these. Our study was not designed to measure any effects from iron that may cause changes to intestinal function unrelated to microbiota reprofiling (for example, impaired barrier function), which may also cause diarrhea. Furthermore, antibiotic use was common this population and an analysis of

the association between antibiotic use and the microbiome, including the interaction with iron interventions, has been previously presented[27]. Finally, although efforts were taken to minimize environmental contamination of samples, this remains a potential limitation of this field-based study in rural Bangladesh.

Ultimately, although our results do not definitively confirm that iron interventions adversely reprofile the infant gut microbiome, an adverse effect of iron on the gut microbiome remains biologically plausible. Thus, the risk-benefit of iron should be carefully considered before implementing these interventions.

## Methods

### Inclusion and ethics statement

The BRISC trial was conceived within a long standing partnership between the International Center for Diarrheal Disease Research, Bangladesh (icddr,b), The Walter and Eliza Hall Institute of Medical Research (WEHI, Australia), and The Peter Doherty Institute, University of Melbourne (Australia). icddr,b, WEHI and Doherty Institute researchers co-developed the study design and methods and were named co-investigators on research grants. Local health workers were involved in design of study procedures. Through the BRISC trial program, one Bangladeshi student has undertaken a PhD program at WEHI in Australia.

The trial protocol, including stool collection and microbiome analyses, was approved by ethics committees at the icddr,b and Melbourne Health, Melbourne, Australia, and overseen by an independent Data Safety and Monitoring Board. The full protocol is available with the main BRISC outcome publication, and sub-study methods have been published previously.[12,27] Parents or guardians of participants gave written informed consent prior to enrollment and were compensated for travel costs for visits.

### Study design

This microbiome sub-study recruited the final 1093 children from the BRISC trial (ACTRN12617000660381), set in rural parts of Rupganj Upazila, Bangladesh. Briefly, BRISC was a three-arm, individually randomized, placebo-controlled trial that recruited 3300 children aged eight months and randomized them 1:1:1 to receive one of iron syrup (and placebo MNPs); iron, zinc, ascorbic acid, vitamin A and folate-containing MNPs (and placebo syrup) or placebo (placebo syrup and placebo MNPs) daily for three months, reflecting WHO guidelines.[4,5] Iron interventions were 12.5 mg ferrous sulphate in syrup and 12.5 mg ferrous fumarate in MNPs. Individual block randomization method was used, with stratification by sex and region (Union) to maintain balance between intervention arms. Blinding was achieved through the use of identical packaging that carried the randomization code for each participant. All participants were required to take two interventions each day: a syrup and a powder. Participants in the trial attended major visits at three time points: baseline – 8 months of age; midline – 11 months of age, immediately following completion of the intervention; and endline – 20 months of age. Data regarding daily adherence to the intervention and parent-reported diarrhea incidence (using the WHO definition of ≥3 loose or liquid stools per day[28]) was obtained by field staff during weekly visits over the 3-month intervention period. The primary outcome was child development after intervention, with a range of other developmental, anthropometric and laboratory outcomes also measured.[29] For this sub-study, we aimed to include the final 1000 participants randomized during the final 6 months of recruitment.

Samples were collected at baseline, midline and endline visits. We sought to perform 16S rRNA gene amplicon sequencing from all stool samples. This allows analysis of microbiome alpha and beta diversity, and taxonomic profiling at the genus level. Additionally, in approximately one-third of samples, we performed shotgun metagenomic sequencing (with priority given where canonical baseline, midline and

endline samples were available), enabling us to further interrogate composition to species level and to perform functional analysis. All sample collection, processing, and bioinformatic and statistical data analysis were planned while blinded to the intervention arm.

## Stool collection and transport

At recruitment, guardians of sub-study participants were given instructions for collection of stool samples, and nappies and specimen containers were provided. Guardians were asked to collect a sample of stool passed within 3 h of the visit. Study staff were sent to retrieve the samples if stool was passed after a visit. Once provided to the field team, samples were stored on ice and transported to the central laboratory in the field within 3 h. There, aliquots were made, DNA/RNA Shield (Zymo Research) was added and aliquots were stored at −20 °C. Specimens were then transported on dry ice to a −80 °C freezer at the International Centre for Diarrheal Disease Research, Bangladesh (icddr,b) facility in Dhaka, Bangladesh, then shipped on dry ice to The Walter and Eliza Hall Institute of Medical Research (WEHI), Australia.

## Laboratory measures

Venous blood samples were obtained at baseline, midline and endline visits. Hemoglobin was measured in the study field office (HemoCue 301+), and ferritin and C-reactive protein (CRP) were measured at icddr,b. Remaining sample aliquots were transported frozen to WEHI, Melbourne, where hepcidin ELISA assays (Intrinsic Life Sciences) were performed.

## DNA extraction from stool for microbiome analysis

Aliquots of samples were added to PowerBead Pro bead tubes along with lysis buffer (Solution CD1 from DNeasy PowerSoil Pro Kit, comprising sodium thiocyanate >= 1 - <10% w/w) and were homogenized on a TissueLyser LT (Qiagen, Venlo, Netherlands) for 10 min at maximum speed. DNA was then extracted with the DNeasy PowerSoil Pro Kit (Qiagen, Venlo, Netherlands) according to manufacturer protocol.

## 16S rRNA gene amplicon sequencing

The V4 hypervariable region of the 16S rRNA gene was targeted using universal primers (Integrated DNA Technologies) in an initial PCR reaction (*Supplementary Material*). Next, dual-index barcodes (forward and reverse) were introduced in a second PCR reaction to join the common overhang sequences in the first reaction to enable sample identification during analysis. This 16S rRNA amplicon sequencing method was developed by the WEHI genomics core facility and has been used successfully in previous applications.[30] PCR plates each contained at least one positive control (a healthy donor stool that underwent DNA extraction in parallel with sub-study samples), one negative DNA extraction control and several PCR negative controls (blanks). Libraries, including negative and positive controls, were sequenced at the WEHI genomics core facility on a MiSeq instrument (Illumina, San Diego, USA) using a 300 bp paired-end protocol with 600 cycles.

## Shotgun metagenomic sequencing

A subset of samples and controls underwent shotgun metagenomic sequencing using the QIAseq FX DNA Library Kit (Qiagen, Venlo, Netherlands), with input DNA and reagents used at 0·5x the specified volumes. DNA from samples was first enzymatically fragmented for eight minutes to achieve a fragment size of 450 base pairs. Adapter ligation was then performed, followed by 8 cycles of PCR amplification. Qubit™ dsDNA Assay kit (Thermo Fisher Cat#Q32851) was used to quantify library concentrations, and library size was calculated using D1000 ScreenTape (Agilent Cat# 5067-5582) in the TapeStation 4200 (Agilent Cat# G2991BA). Equimolar amounts of libraries were pooled and sequenced using NovaSeq (Illumina, San Diego, USA).

## Bioinformatic and statistical analysis

16S rRNA amplicon sequence files were processed with the *DADA2* package in RStudio.[31] Data generated from this pipeline were used to create an object in *phyloseq*.[32] Samples with <500 reads and amplicon sequence variants (ASVs) with zero reads were removed, and ASVs were agglomerated at the genus level before further analysis. Alpha diversity analysis used the *microbiomeSeq* package in RStudio using Shannon and inverse Simpson indices.[33] Group differences between trial arms were calculated using pair-wise analysis of variance (ANOVA) for each diversity measure. Beta diversity calculation used *phyloseq* to calculate and plot Bray-Curtis dissimilarity. *Vegan* was used to calculate numerical values and significance testing.

*ANCOM-BC* was used with default settings to calculate differential abundance at the genus level. Where an adjusted *p*-value is shown, this was calculated using the Benjamini-Hochberg method to control for the FDR at a level of 5%.[34] Taxa were also required not to be sensitive to pseudo-count addition in order to be considered differentially abundant.[35]

Our primary analyses applied correction for FDR. However, because this study evaluates a safety outcome for which false positive and false negative findings could have important public health impacts, we also explored and transparently reported findings based on the unadjusted *p*-value. This aligns with the conservative approach of the main trial in which efficacy outcomes were adjusted for multiple testing, but safety outcomes were reported without adjustment.[12,23]

Shotgun metagenomic samples were processed using the *bioBakery workflows* tool, comprising the default quality control steps in *KneadData* and taxonomic classification in *MetaPhlAn*.[36] Taxonomic differential abundance by trial metadata was performed at the species level using *MaAsLin2* with a minimum species prevalence cut-off of 10% of samples.[37] Centered log ratio normalization/transformation was performed on relative abundance data in *MaAsLin2*, with outputs for differentially abundant species including coefficient (approximating $log_2$-fold change), standard deviation, and adjusted *p*-values (using Benjamini-Hochberg method with an FDR cut-off of 0·05).[34] As above, our primary analyses applied adjustment for false discovery; in a further analysis, we explored and transparently report unadjusted effects.

Competitive phylum set enrichment analysis was conducted for two phyla previously reported to be impacted by iron interventions,[8,9] using *cameraPR* from the R package *limma* (version 3.56.2), using t-statistics for the ranking of species.[38,39] This approach evaluates the rank order of expression intensity for species within a specific subset in contrast to all other species. It examines whether there are statistically significant variations in the aggregate mean rank order for the entire set between different treatment conditions, thus providing insight into the differential expression patterns induced by the treatments.

Microbial functional pathways were profiled using *HUMAnN* in the *bioBakery workflow*s using *ChocoPhlAn UniRef90*, *MetaCyc* and *MinPath* databases.[36,40–42] The resulting pathway abundance files were merged, unstratified from corresponding organisms, and renormalized from reads per kilobase (RPKs) to relative abundances. *HUMAnN* outputs were then processed using *MaAsLin2* as above to calculate differential abundance at the unstratified pathway level according to trial arm, adjusted for multiple comparisons using Benjamini-Hochberg method at a level of 5%.

## Reporting summary

Further information on research design is available in the Nature Portfolio Reporting Summary linked to this article.

## Data availability

The sequencing data used in this study have been deposited in the NCBI Sequence Read Archive (SRA) database under BioProject accession PRJNA1081952 [https://www.ncbi.nlm.nih.gov/sra]. Source data are provided with this paper.

## Code availability

Code used to process sequencing files (in R for 16S rRNA files and using Python for shotgun metagenomic files), perform statistical analysis and generate figures (in R) is based on the respective packages cited in the manuscript. Custom code was not used. However, code is available from the authors by request.

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

## Acknowledgements

We thank the local field workers for their support, and all participants and their families for their involvement in the study. The authors gratefully acknowledge the WEHI Advanced Genomics Facility for their support and assistance in this work. This work was supported by the Australian National Health and Medical Research Council GNT1103262 (B.A.B.), GNT1159151 (S.R.P.), GNT1158696 (S.R.P.) and GNT2009047 (S.R.P.), and by The Geok Hua Wong Charitable Trust. icddr,b is also grateful to the Governments of Bangladesh, Canada, Sweden, and the UK for providing core/unrestricted support. This work was made possible through Victorian State Government Operational Infrastructure Support and Australian Government NHMRC IRIISS. AB was supported by a Research Training Program Scholarship from the University of Melbourne and a stipend from WEHI. The study funder had no role in study design, data collection and analysis or manuscript writing.

## Author contributions

A.B. designed the study, led the fieldwork, performed the microbiome experiments, undertook bioinformatic data analysis, interpreted the findings, and wrote the manuscript. S.B. designed the study, undertook statistical analysis, and wrote the manuscript. M.I.H. and J.H. designed the study, led the fieldwork, and interpreted the findings. M.B., N.J., S.W., and R.B. assisted in designing and performing the microbiome experiments. A.J., I.A.A., and P.K.R. assisted with bioinformatic data analysis. L.M.L., C.B., R.A., D.C., M.Z., and A.J. interpreted the findings. M.S.A.B. assisted with fieldwork. B.A.B. designed the study and the parent trial. S.R.P. designed the study and parent trial, interpreted the findings, and wrote the manuscript. All authors reviewed and approved the final version of the manuscript.

## Competing interests

The authors declare no competing interests.

## Additional information

[1]Population Health and Immunity Division, Walter and Eliza Hall Institute of Medical Research, Parkville, VIC, Australia. [2]Department of Medical Biology, The University of Melbourne, Parkville, VIC, Australia. [3]Centre for Epidemiology and Biostatistics, University of Melbourne School of Population and Global Health, Carlton, Carlton, VIC, Australia. [4]Department of Infectious Diseases at the Peter Doherty Institute of Infection and Immunity, The University of Melbourne, Melbourne, VIC, Australia. [5]International Center for Diarrheal Diseases Research, Bangladesh (icddr,b), Dhaka, Bangladesh. [6]Advanced Technology and Biology Division, Walter and Eliza Hall Institute of Medical Research, Parkville, VIC, Australia. [7]Faculty of Science, University of Melbourne, Melbourne, VIC, Australia. [8]Diagnostic Haematology, The Royal Melbourne Hospital, Parkville, VIC, Australia. [9]Department of Health Promotion, Education, and Behavior, Arnold School of Public Health, University of South Carolina, Columbia, SC, USA. [10]Medical Research Council Translational Immune Discovery Unit, MRC Weatherall Institute of Molecular Medicine, John Radcliffe Hospital, University of Oxford, England, UK. [11]Victorian Infectious Diseases Service, Royal Melbourne Hospital, Parkville, VIC, Australia. [12]Clinical Haematology at The Royal Melbourne Hospital and the Peter MacCallum Cancer Centre, Parkville, VIC, Australia. ✉e-mail: Baldi.a@wehi.edu.au; Pasricha.s@wehi.edu.au

