## [Peer Review File · Nature Communications]

REVIEWER COMMENTS

Reviewer #1 (Remarks to the Author):

Study Summary: This is a large randomized controlled trial of three arms, iron drop, MNP, and placebo, to evaluate the effects of iron supplementation in two different methods of administration on the gut microbiome. The results show no statistically significant differences among the three groups with adjustment for multiple comparisons but suggest potential unfavorable effects of iron supplementation on Bifidobacterium, Lactobacillus, and Clostridium species especially in iron replete infants. Overall, the study was well conducted and described, low risks of biases, and with reasonable conclusions.

Strengths:

1. This is a major study to clarify the effects of enteral iron supplement on gut microbiota. With a large sample size, it will have large impact on public health in term of universal implementation of iron supplementation.
2. The study's Methods, sample processing, and data analyses were clearly described.
3. The authors clearly described outcomes, strengths, and weaknesses of the study. The conclusions are supported by the results.

Weaknesses:

1. Study design:

- a. The randomization procedure was not described.
- b. The dose assignment was blinded to the analysts but not to parents and study field staff, which can introduce biases. A discussion of potential biases this may introduce is recommended.

2. Methods:

- a. The actual iron dose from the drops and MNP and the doses of other components in the MNP were not given. The effects of iron on gut microbiota maybe dose dependent, so this information will give the readers a sense of how much iron was given.
- b. Parental home stool collection may not be avoidable in this study, but it may introduce environmental influences on the stool microbiota prior to processing. It could be up to 6 hours or more from collection to freezing. How would this affect the microbiome results?
- c. Batch effects in stool microbiome sequencing and analyses need to be addressed at the sequencing level and at the bioinformatic level if possible.

3. Minors:

- a. Figure 5 D-F and SF3D are hard to see with all the labels overlapping with the dots. Perhaps the dots can be labeled with numbers and add a legend to explain the numbers.
- b. How was diarrheal disease in 8–20-month-old children defined for parents to report? Children at this age have different stool consistencies which makes it hard to clinically diagnose true pathologic diarrheal conditions.
- c. Any data on antibiotic exposure? This can have effects on “diarrheal disease” occurrence and changes in gut microbiota.
- d. Children 8 months to 11 months are quickly exposed to new foods, new social and environmental exposures, which can affect the gut microbiota development. What were these exposures in studied children among the groups?

Reviewer #2 (Remarks to the Author):

Baldi and colleagues present an interesting work aimed at surveying the effect of iron supplementation on the gut microbiome of more than 900 Bangladeshi infants as part of the universal iron supplementation recommendations by WHO in settings where anaemia is prevalent. They compare their results with other previous smaller studies which highlighted the increase in pathogens and depletion of commensals in children subjected to iron supplementation. Baldi and colleagues present their findings with both correcting for FDR and unadjusted to be able to detect even minor changes in the microbiome structure. They cross-check their unadjusted findings with biologically relevant observations, therefore justifying this approach and with the key aim of ensuring safety for infants. I enjoyed reading this study, as I think it is a very important and well-done work.

Nevertheless, some important points need to be fixed before publication, which I report below.

1. Data availability. The results are presented to the reader in the text, with no supplementary table reporting the taxonomic assignments and relative abundances for each taxon at the different time points. The same is true for pathways analysis, where only a subset of pathways are reported in the figures without info on their abundance. Unless the reader downloads the data from SRA (not available yet) and re-analyses the whole cohort, there is no way to check that what is reported in the text is correct and its real implication besides what the Authors present in the text. This is particularly relevant in cases where minor changes in low abundance taxa/pathways may result in significant results according to standard analysis methods (such as HUMAnN) but no real impact on the overall microbiome. I therefore recommend the Authors to please add the following supplementary items:

- A) Taxa table for 16S data, with relative abundances for all samples at all time points
- B) Taxa table for shotgun data, with relative abundances for all samples at all time points
- C) Pathways / Functional table

2. Microbial nomenclature. Please use the new phyla names approved by the International Code of Nomenclature of Prokaryotes (Line 278 and others), and if you wish to keep the old names as a reference use them in brackets at the first occurrence. Additionally, make sure that the same coherent nomenclature is used throughout the text, figures, and tables, as for instance in Figures 1D and 1E different names are used for the same phylum (Actibacteriota/Actinobacteria, etc). Also a minor comment on this, please check italics for genera and species throughout the text (e.g. Line 95).

3. Discussion section. The paper is very well written, except for the discussion section which lacks a coherent thread/story. Some parts are repeated (e.g. the rationale behind adjusted/unadjusted results reporting) and there is no real discussion of the reported results except for 418-433 and 443-446. From 448 onward there is a long list of strengths of the study (which I support) but then there are only a few lines on the limitations, despite these limitations being quite big (i.e. lack of eukaryotic/viral community information) and important given that viral infections are the leading cause of diarrhoea in Bangladeshi children, as the Authors write in the text. In general, the discussion should be remodelled

to remove unnecessary repetitions regarding the study design (also detailed in the methods), rebalance the strengths vs limitation section (authors don't need to convince anyone of the importance of the study, it is well done and this section on the strengths is quite unnecessary), to further discuss how these limitations may be overcome by future studies by the authors or others (e.g. discuss the importance of viral infections and how this dataset of shotgun seq can be leveraged to further assess this component, for instance by naming relevant DNA viruses causing diseases in the intervention population?) and by putting more details when comparing with other studies (e.g. individuals/samples nr, outcomes etc)

I also have some minor comment:

- Line 112: "In a large sub-study": is it the one presented here or the larger microbiome study including other projects? Please clarify this sentence.
- Line 171: which lysis buffer? Please specify the composition or the commercial name, if a commercial buffer has been used.
- Lines 177-181: please briefly report here the PCRs conditions and primers sequences to help readers willing to replicate results
- Line 274: please provide here the link to the supplementary table containing the genus level taxonomic table
- Lines 283-289: please provide the raw data for pathway abundance pre- and post-intervention as Supplementary table. How abundant were these pathways wrt the average? At low abundances, some very minor changes may result in statistically significant results but they may be due to chance.
- Lines 295-296: please provide a supplementary table with the taxonomic assignment and abundances for both 16S and shotgun data and link it/them here
- Lines 394 and others: please refrain from using "gut flora" when referring to the microbiome / microbial communities in the gut
- Lines 401-403: please clarify the meaning of this sentence. Do the Authors mean that shotgun sequencing provides stronger statistical power to the results? If so, I'm not sure this is the case, maybe the Authors may want to highlight the power of shotgun sequencing to look at species and be able to tell apart commensals from pathogens clearly.
- Lines 419-421: please provide some bibliographic support for this statement
- Lines 445-446: so what are the Authors suggesting as future steps? Please better explain why a screen-and-treat program would be more complex and whether this should be considered given the potential safety risks for iron-repleted children. Are there other LMICs where a screen-and-treat program is in place?
- Lines 467-468: I guess here the Authors meant that the next step would be to evaluate the non-bacterial fraction of the community? I don't see why the research should only focus on non-bacterial DNA pathogens, unless they mean that their dataset of DNA seq can be leveraged to look at the DNA viruses and fungi? Or that they already have a study coming out with these results? Please clarify this sentence.
- Acknowledgments: no fundings are reported.
- Data availability: when will data be public?

Reviewer #3 (Remarks to the Author):

This is an interesting manuscript that leverages findings from a large RCT comparing iron supplementation with/without MNPs to placebo in a cohort of Bangladeshi children to evaluate changes in the microbiome associated with the intervention. The study adds some additional data to existing studies that have examined these associations, particularly in relation to concerns that iron may negatively impact the microbiome and predispose infants to pathogen growth and colonization.

There are a number of limitations to this analysis that limit the overall impact of this study. Before discussing those specific issues, I believe there is a fundamental issue with authorship that must be clearly addressed. It appears that the trial was conducted in Bangladesh and a number of Bangladeshi authors contributed to the study design, implementation and write up. It is not clear why both the last and first authors are from high income institutions and why a Bangladeshi author is not acknowledged in a similar authorship position. Unless appropriate justification can be provided, it is important to ensure that colleagues in LMIC settings are given equal opportunity for first and last author positions – and given the apparent location and implementation needs of this study, this needs to be clearly explained.

The study has several key limitations;

- 1) This particular analysis did not focus on clinical outcomes – instead evaluated proxy measures of potential harm. As such, I do not think safety is appropriate to reference here. This is a study of potential adverse impact.
- 2) Not all diarrhea is due to infectious causes. Without assessment of functional attributes (permeability, barrier function, etc.) it is not possible to rule out adverse impacts on the gut from these interventions.
- 3) It appears that selection of these children was based on the presence of follow up samples. This is hugely problematic and completely eliminates the randomization benefits of the study as well as introducing sampling bias. In fact, there appear to be meaningful differences in baseline characteristics between the arms (iron deficiency at baseline for example). This is likely the result of sampling bias.
- 4) Retention in the study was very poor which is problematic as it may have led to significant bias in the presented outcomes. Children who completed the study may be fundamentally different from those who did not. In addition, the benefits of randomization are attenuated when there is differential loss to follow up. Please present data comparing baseline characteristics among those who completed follow/did not complete follow up in each arm.
- 5) Adherence to the intervention was also problematic. First, adherence is not a baseline feature so should not be presented in Table 1. Second, the denominator to assess adherence is not the proportion of children taking the intervention who were retained in the study – it is the proportion of all enrolled who took the intervention. When applying this, adherence was quite low. As a result, it is very difficult to determine the true impact of the intervention in this study. In addition, the poor adherence is complicated by the timing of the sampling. If many children were not adherent, then you are comparing children who received the intervention sooner vs. later to the outcome assessment. While the authors did compare outcomes among children fully adherent and those not adherent, this is likely limited by power do detect such differences.
- 6) The findings of reduced Bifidobacterium and Lactobacillus spp. are important given potential consequences for growth and sepsis. This should be discussed.
- 7) Another important potential concern is the propagation of AMR genes as a result of the intervention.

This would be a valuable addition.

8) The overall conclusions of the paper are not strong and could be revised.

Additional minor comments below;

1) Line 66 – it is not clear what the authors mean by “diarrheal disease risk”.

2) Line 382 – the authors state that these changes are not due to false discovery. This cannot be determined definitively and this statement should be modified – it is overstated.

REVIEWER COMMENTS

Reviewer #1 (Remarks to the Author):

Study Summary: This is a large randomized controlled trial of three arms, iron drop, MNP, and placebo, to evaluate the effects of iron supplementation in two different methods of administration on the gut microbiome. The results show no statistically significant differences among the three groups with adjustment for multiple comparisons but suggest potential unfavorable effects of iron supplementation on Bifidobacterium, Lactobacillus, and Clostridium species especially in iron replete infants. Overall, the study was well conducted and described, low risks of biases, and with reasonable conclusions.

We thank the reviewer for their assessment of our manuscript and have addressed the specific comments in the table below.

REVIEWER COMMENT	AUTHOR RESPONSE	MANUSCRIPT REFERENCE
Strengths: 1. This is a major study to clarify the effects of enteral iron supplement on gut microbiota. With a large sample size, it will have large impact on public health in term of universal implementation of iron supplementation. 2. The study's Methods, sample processing, and	We appreciate the positive feedback from the reviewer.	

REVIEWER COMMENT	AUTHOR RESPONSE	MANUSCRIPT REFERENCE
data analyses were clearly described. 3. The authors clearly described outcomes, strengths, and weaknesses of the study. The conclusions are supported by the results.		
Weaknesses: 1. Study design: a. The randomization procedure was not described.	Thank you for this comment. This sub study comprised participants recruited from the main BRISC iron trial, and as such the randomization was the same. BRISC used individual block randomization, with stratification by sex and sub-centre to maintain balance between intervention arms. We have added the following text to the Methods section: 'Individual block randomization method was used, with stratification by sex and region (Union) to maintain balance between intervention arms.'	Line 173
b. The dose assignment was blinded to the analysts but not to parents and study field staff, which can introduce biases.	Thank you for this comment. Note that the study was blinded through the use of placebo in a double dummy fashion, ensuring participants/families, all study staff and statisticians were unaware of allocation until formal unblinding, minimizing the risk of bias. This is strength of our study design. We have inserted the below explanation (from the BRISC study protocol) into the Methods section.	

REVIEWER COMMENT	AUTHOR RESPONSE	MANUSCRIPT REFERENCE
A discussion of potential biases this may introduce is recommended.	'Blinding was achieved through the use of identical packaging that carried the randomization code for each participant. All participants were required to take two interventions each day: a syrup and a powder.' (hence the trial was double-dummy, double-blinded) With regard to this and the above comment, we have also included in the Methods section references to both the BRISC trial publication and its associated statistical analysis plan. 'The full protocol is available with the main BRISC outcome publication.'	Line 175 Line 158
2. Methods: a. The actual iron dose from the drops and MNP and the doses of other components in the MNP were not given. The effects of iron on gut microbiota maybe dose dependent, so this information will give the readers a sense of how much iron was given.	We thank you for drawing our attention to this omission. The iron dose in both formulations was identical and we have added this to the Methods section (as below): 'Iron interventions were 12.5mg ferrous sulphate in syrup and 12.5mg ferrous fumarate in MNPs.'	Line 169

REVIEWER COMMENT	AUTHOR RESPONSE	MANUSCRIPT REFERENCE
b. Parental home stool collection may not be avoidable in this study, but it may introduce environmental influences on the stool microbiota prior to processing. It could be up to 6 hours or more from collection to freezing. How would this affect the microbiome results?	Thank you for this comment. Obtaining biological samples in a low-resource setting does indeed present challenges for research. We endeavoured to minimize any effects of environmental contamination by 1. Giving guardians clear verbal and written (with illustrations) instructions about clean stool collection, 2. Equipping guardians with gloves and specimen jar, and 3. Instructing guardians to call the field worker as soon as the stool had been collected (or instructing them to deliver it immediately to the field centre for processing). We also sought to reduce contamination and degradation of the samples between collection and freezing by minimizing the time from collection to frozen storage. We instructed laboratory workers to process samples as they arrived, which involved splitting the sample into two aliquots under aseptic conditions and immediately storing them in the -20°C freezer. Samples were then transported in batches to a -80°C freezer. We have added the following text to the Discussion: 'Finally, although efforts were taken to minimize environmental contamination of samples, this remains a potential limitation of this field-based study in rural Bangladesh.'	Line 582
c. Batch effects in stool microbiome sequencing and analyses need to be addressed at the sequencing level and at the bioinformatic level if possible.	We thank the reviewer for this comment. In the Reporting Summary we have outlined our approach to sequencing to minimize bias: 'Samples were also sequenced in random order (i.e. PCR plates and sequencing runs included samples from different timepoints) to minimize any batch effects.' This applies to both 16S amplicon and shotgun metagenomic sequencing. Read counts from all sequencing runs were adequate (see first table below), with the vast majority of processed 16S fastq files yielding reads of ≥500 regardless of sequencing run. The	Reporting Summary

REVIEWER COMMENT	AUTHOR RESPONSE	MANUSCRIPT REFERENCE																								
	bioinformatic pipeline was performed on all samples at the same time (after all sequencing had been performed).    16 sequencing run Reads (median) Reads (standard deviation)     1 18296 5842   2 24619 7961   3 5515 3781   4 17111 13266   5 9315 5200   6 16427 6495   7 13345 4789    We also included at least one negative and one positive control (healthy donor stool) on each PCR plate and, for shotgun PCR plates we also included a commercial community standard. To illustrate, each of the three NovaSeq runs for shotgun metagenomic analysis comprised four PCR plates. Negative controls (table below) show either no final reads after KneadData processing, and those with a few reads yielded no results mapped to taxonomy (aside from reads from Plate 1.4 with reads corresponding to B. longum).	16 sequencing run	Reads (median)	Reads (standard deviation)	1	18296	5842	2	24619	7961	3	5515	3781	4	17111	13266	5	9315	5200	6	16427	6495	7	13345	4789	
16 sequencing run	Reads (median)	Reads (standard deviation)																								
1	18296	5842																								
2	24619	7961																								
3	5515	3781																								
4	17111	13266																								
5	9315	5200																								
6	16427	6495																								
7	13345	4789																								

REVIEWER COMMENT	AUTHOR RESPONSE	MANUSCRIPT REFERENCE																																																				
	   Negative control sample Final reads (forward) Final reads (reverse) Taxonomy     Plate 1.1 NA NA    Plate 1.2 NA NA    Plate 1.3 NA NA    Plate 1.4 4664 4664 100% mapped reads Bifidobacterium longum   Plate 2.1 NA NA    Plate 2.2 14 14 Nil mapped   Plate 2.3 NA NA    Plate 2.4 46 46 Nil mapped   Plate 3.1 NA NA    Plate 3.2 451 451 Nil mapped   Plate 3.3 NA NA    Plate 3.4 7 7 Nil mapped    Between the two types of positive control there was similar taxonomic profiling at the genus level of samples of the same origin (figure below), and while the community standard in Plate 1.2 did not yield the expected taxonomic composition, the corresponding healthy donor sample on the same plate showed appropriate composition compared with the same control on the other plates. Randomization of samples across plates and running of negative and two forms of positive control across every plate helps ensure the quality of our study.	Negative control sample	Final reads (forward)	Final reads (reverse)	Taxonomy	Plate 1.1	NA	NA		Plate 1.2	NA	NA		Plate 1.3	NA	NA		Plate 1.4	4664	4664	100% mapped reads Bifidobacterium longum	Plate 2.1	NA	NA		Plate 2.2	14	14	Nil mapped	Plate 2.3	NA	NA		Plate 2.4	46	46	Nil mapped	Plate 3.1	NA	NA		Plate 3.2	451	451	Nil mapped	Plate 3.3	NA	NA		Plate 3.4	7	7	Nil mapped	
Negative control sample	Final reads (forward)	Final reads (reverse)	Taxonomy																																																			
Plate 1.1	NA	NA																																																				
Plate 1.2	NA	NA																																																				
Plate 1.3	NA	NA																																																				
Plate 1.4	4664	4664	100% mapped reads Bifidobacterium longum																																																			
Plate 2.1	NA	NA																																																				
Plate 2.2	14	14	Nil mapped																																																			
Plate 2.3	NA	NA																																																				
Plate 2.4	46	46	Nil mapped																																																			
Plate 3.1	NA	NA																																																				
Plate 3.2	451	451	Nil mapped																																																			
Plate 3.3	NA	NA																																																				
Plate 3.4	7	7	Nil mapped																																																			

REVIEWER COMMENT	AUTHOR RESPONSE	MANUSCRIPT REFERENCE
	 Taxonomic Bar Chart of Shotgun Community Standards and Positive Control Samples The chart displays the relative abundance (%) of various bacterial genera across two groups of samples: Community standard (Plates 1.1-3.4) and Positive control (Plates 1.1-3.4). The y-axis represents relative abundance from 0 to 100%. The legend includes 20 genera: Bifidobacterium, Lactobacillus, Streptococcus, Staphylococcus, Enterococcus, Listeria, Escherichia, Blautia, Bacillus, Salmonella, Lachnospiraceae, Faecalibacterium, Roseburia, Fusicatenibacter, Collinsella, Pseudomonas, Anaerostipes, Bacteroides, Eubacterium, and Klebsiella.	
3. Minors: a. Figure 5 D-F and SF3D are hard to see with all the labels overlapping with the dots. Perhaps the dots can be labeled with numbers and add	Thank you for this feedback. We have revised those figures accordingly.	Figure 5D-F Figure S3D

REVIEWER COMMENT	AUTHOR RESPONSE	MANUSCRIPT REFERENCE
a legend to explain the numbers.		
b. How was diarrheal disease in 8–20-month-old children defined for parents to report? Children at this age have different stool consistencies which makes it hard to clinically diagnose true pathologic diarrheal conditions.	Thank for you for this comment. Field workers used the World Health Organization definition of diarrhoea in weekly and monthly visits to obtain parent-reported information. For clarity we have moved the WHO definition of diarrhea to the Methods section and removed it from the Results section. The revised part of the Methods section now reads: 'Data regarding daily adherence to the intervention and parent-reported diarrhea incidence (using the WHO definition of ≥ 3 loose or liquid stools per day) was obtained by field staff during weekly visits over the 3-month intervention period.'	Line 180
c. Any data on antibiotic exposure? This can have effects on “diarrheal disease” occurrence and changes in gut microbiota.	Data was collected during weekly (8-11 months) and monthly (11-20 months) visits regarding incidence of infection and medication use, including use of antibiotics. Antibiotic use was associated with transient changes in microbiome composition, and a manuscript describing these findings is currently in press (provisionally accepted) at Nature Communications. In that manuscript we present an analysis that demonstrated that there was no evidence of interaction between study arm (iron/MNPs or placebo) and the effect of antibiotic use on microbiome composition (see below). As these data will be published elsewhere we will not	

REVIEWER COMMENT	AUTHOR RESPONSE	MANUSCRIPT REFERENCE
	include them in this manuscript, but we have reproduced two figures below. The first is a volcano plot illustrating the interaction between BRISC trial arm (iron/MNPs and placebo) and antibiotic use in the preceding 7 days in relation to differential abundance of AMR genes, and the second in relation to taxonomic differential abundance at the genus level.  The figure is a volcano plot with the following characteristics: X-axis: \log_2-fold change, ranging from -2.0 to 2.0 with major ticks every 1.0 unit.Y-axis: $-\log_{10}(\text{FDR-adjusted p value})$, ranging from 0.0 to 4.0 with major ticks every 0.5 units.Significance Threshold: A horizontal red line is drawn at approximately $y = 1.3$.Data Points: Numerous small black dots are scattered across the plot. Most are located below the red line, indicating they are not statistically significant. One point is located at approximately $(-0.8, 0.3)$, which is above the red line, indicating it is statistically significant.	

REVIEWER COMMENT	AUTHOR RESPONSE	MANUSCRIPT REFERENCE
	 We have added the following sentence to the Discussion section: 'Furthermore, antibiotic use was common this population and an analysis of the association between antibiotic use and the microbiome, including the interaction with iron interventions, has been previously presented (reference – our publication in press).'	Line 579

REVIEWER COMMENT	AUTHOR RESPONSE	MANUSCRIPT REFERENCE
d. Children 8 months to 11 months are quickly exposed to new foods, new social and environmental exposures, which can affect the gut microbiota development. What were these exposures in studied children among the groups?	The reviewer makes a valuable point, and we acknowledge that many social and environmental variables are known to act on and interact with the gut microbiome. We collected additional data regarding food security, age of introduction of complementary food and present these in Table 1 of the manuscript, and note these are similar between trial arms. Surveys regarding dietary diversity and water, sanitation and hygiene (WASH) were implemented in the sub study population at later time points (midline for WASH, and midline and endline for dietary diversity), and are hence not presented in Table 1. Dietary diversity assessment was a seven-question survey based on food intake over the preceding 24 hours:  1. Grains, roots, and tubers 2. Legumes and nuts 3. Dairy products 4. Flesh foods 5. Eggs 6. Vitamin A-rich fruits and vegetables 7. Other fruits and vegetables We did not perform subgroup analyses based on responses to individual questions or the total score, but we note this as an important example of an environmental influence on the gut microbiome. A summary of total scores (out of seven) at midline is shown in the figure below:	

REVIEWER COMMENT	AUTHOR RESPONSE	MANUSCRIPT REFERENCE																		
	Percentage (%)  Dietary Diversity ScorePercentage (%)11.524.5314.542352261671583.5	Dietary Diversity Score	Percentage (%)	1	1.5	2	4.5	3	14.5	4	23	5	22	6	16	7	15	8	3.5	
Dietary Diversity Score	Percentage (%)																			
1	1.5																			
2	4.5																			
3	14.5																			
4	23																			
5	22																			
6	16																			
7	15																			
8	3.5																			

Reviewer #2 (Remarks to the Author):

Baldi and colleagues present an interesting work aimed at surveying the effect of iron supplementation on the gut microbiome of more than 900 Bangladeshi infants as part of the universal iron supplementation recommendations by WHO in settings where anaemia is prevalent. They compare their results with other previous smaller studies which highlighted the increase in pathogens and depletion of commensals in children subjected to iron supplementation. Baldi and colleagues present their findings with both correcting for FDR and unadjusted to be able to detect even minor changes in the microbiome structure. They cross-check their unadjusted findings with biologically relevant observations, therefore justifying this approach and with the key aim of ensuring safety for infants. I enjoyed reading this study, as I think it is a very important and well-done work.

Nevertheless, some important points need to be fixed before publication, which I report below.

Again we thank the reviewer for their feedback and we address specific comments below.

REVIEWER COMMENT	AUTHOR RESPONSE	MANUSCRIPT REFERENCE
1. Data availability. The results are presented to the reader in the text, with no supplementary table reporting the taxonomic assignments and relative abundances for each taxon at the different time points. The same is true for pathways analysis, where only a subset of pathways are reported in the figures without info on their abundance. Unless the reader downloads the data from SRA (not available yet) and re-analyses the whole cohort, there is no way to check that what is reported in the text is correct and its real implication besides what the Authors present in the text. This is particularly relevant in cases where minor changes in low abundance taxa/pathways may result in	We thank the reviewer for this feedback. To address this we have added a Source Data file to our submission that includes the data used to generate the relevant figures. We have added references to this file in the relevant figure legends.	Source Data

REVIEWER COMMENT	AUTHOR RESPONSE	MANUSCRIPT REFERENCE
significant results according to standard analysis methods (such as HUMAnN) but no real impact on the overall microbiome. I therefore recommend the Authors to please add the following supplementary items: A) Taxa table for 16S data, with relative abundances for all samples at all time points B) Taxa table for shotgun data, with relative abundances for all samples at all time points C) Pathways / Functional table		
2. Microbial nomenclature. Please use the new phyla names approved by the International Code of Nomenclature of Prokaryotes (Line 278 and others), and if you wish to keep the old names as a reference use them in brackets at the first occurrence. Additionally, make sure that the same coherent nomenclature is used throughout the text, figures, and tables, as for instance in Figures 1D and 1E different names are used for the same phylum (Actibacteriota/Actinobacteria, etc). Also a minor comment on this, please check italics for genera and species throughout the text (e.g. Line 95).	Thank you for this comment. We have revised the manuscript and figure labels (including for Figures 1D-E) to reflect this up-to-date nomenclature.	Lines 338-39, 416-18, 465, 496-97, 500-01 Figures 1D-E
3. Discussion section. The paper is very well written, except for the discussion section which lacks a coherent thread/story. Some parts are repeated (e.g. the rationale behind adjusted/unadjusted results reporting) and	We appreciate this feedback from the reviewer. We have shortened the Discussion section to improve readability and reduce repetition. In particular we have shortened the strengths section, and have discussed the risk of viral diarrhea and future	

REVIEWER COMMENT	AUTHOR RESPONSE	MANUSCRIPT REFERENCE
there is no real discussion of the reported results except for 418-433 and 443-446. From 448 onward there is a long list of strengths of the study (which I support) but then there are only a few lines on the limitations, despite these limitations being quite big (i.e. lack of eukaryotic/viral community information) and important given that viral infections are the leading cause of diarrhoea in Bangladeshi children, as the Authors write in the text. In general, the discussion should be remodelled to remove unnecessary repetitions regarding the study design (also detailed in the methods), rebalance the strengths vs limitation section (authors don't need to convince anyone of the importance of the study, it is well done and this section on the strengths is quite unnecessary), to further discuss how these limitations may be overcome by future studies by the authors or others (e.g. discuss the importance of viral infections and how this dataset of shotgun seq can be leveraged to further assess this component, for instance by naming relevant DNA viruses causing diseases in the intervention population?) and by putting more details when comparing with other studies (e.g. individuals/samples nr, outcomes etc)	opportunities for analyzing these pathogens using the shotgun metagenomics data. We have specifically included the following text in response to the request for examples of viral infections causing diarrhea in this setting: 'Viruses are among the leading causes of diarrheal infections requiring hospitalization in children in Bangladesh, and this is likely the case in many other LMICs.⁴⁰ For example, adenovirus is among the most common causes of diarrhea in Bangladesh in both children and adults.⁴¹ Future analysis will evaluate these.' We have added sample sizes and outcome measurement techniques to our discussion of previous studies.	Line 573
I also have some minor comment:	Thank you for this comment. We have clarified this phrase as below:	

REVIEWER COMMENT	AUTHOR RESPONSE	MANUSCRIPT REFERENCE
- Line 112: "In a large sub-study": is it the one presented here or the larger microbiome study including other projects? Please clarify this sentence.	'In this large sub-study we present here, we evaluated stool samples from a subset of BRISC participants at baseline, after three months of intervention, and after a further 9-month post-intervention follow-up.'	Line 131
- Line 171: which lysis buffer? Please specify the composition or the commercial name, if a commercial buffer has been used.	We have added details in the Methods section regarding the lysis buffer used: 'Aliquots of samples were added to bead tubes (PowerBead Pro) along with lysis buffer (Solution CD1 from DNeasy PowerSoil Pro Kit, comprising sodium thiocyanate ≥ 1 - $<10\%$ w/w) and were homogenized on a TissueLyser LT (Qiagen, Venlo, Netherlands) for 10 minutes at maximum speed.'	Line 227
- Lines 177-181: please briefly report here the PCRs conditions and primers sequences to help readers willing to replicate results	Thank you for this suggestion. We have added details about PCR primers and conditions in the Supplementary Information and made reference to this document in the Methods section.	Line 237 Supplementary Information
- Line 274: please provide here the link to the supplementary table containing the genus level taxonomic table	Thank you. This reference has been added.	Line 335
- Lines 283-289: please provide the raw data for pathway abundance pre- and post-intervention as Supplementary table. How abundant were these pathways wrt the average? At low abundances, some very minor changes may result in statistically significant results but they may be due to chance.	Thank you. This reference has been added.	Line 357

REVIEWER COMMENT	AUTHOR RESPONSE	MANUSCRIPT REFERENCE
- Lines 295-296: please provide a supplementary table with the taxonomic assignment and abundances for both 16S and shotgun data and link it/them here	Thank you. This reference has been added to the figure legend for Figure 3A-B.	Line 367
- Lines 394 and others: please refrain from using “gut flora” when referring to the microbiome / microbial communities in the gut	We have amended these sections accordingly.	Lines 478, 523, 534, 612
- Lines 401-403: please clarify the meaning of this sentence. Do the Authors mean that shotgun sequencing provides stronger statistical power to the results? If so, I'm not sure this is the case, maybe the Authors may want to highlight the power of shotgun sequencing to look at species and be able to tell apart commensals from pathogens clearly.	Thank you for this comment. We have amended this statement to remove reference to statistical power, as below: 'Our 16S rRNA gene sequencing study exceeds the cumulative sample size across these studies; in addition, by applying shotgun metagenomics to this problem, our study provides higher sequencing resolution to address this problem.'	Line 485
- Lines 419-421: please provide some bibliographic support for this statement	Thank you – a reference for this statement has been added.	Line 518
- Lines 445-446: so what are the Authors suggesting as future steps? Please better explain why a screen-and-treat program would be more complex and whether this should be considered given the potential safety risks for iron-repleted children. Are there other LMICs where a screen-and-treat program is in place?	We have added the following sentence to the paragraph: 'However, a screen-and-treat program would require near-patient testing for iron deficiency prior to provision of iron, which would raise the cost and complexity of this public health program and may leave still some anemia untreated. Such cost-effective testing technology remains an unmet need. This approach was explored in two Gambian randomized trials of pre-iron supplementation hepcidin screening in pregnancy and in	Line 540

REVIEWER COMMENT	AUTHOR RESPONSE	MANUSCRIPT REFERENCE
	infants that did not demonstrate non-inferiority in terms of anemia control'.	
- Lines 467-468: I guess here the Authors meant that the next step would be to evaluate the non-bacterial fraction of the community? I don't see why the research should only focus on non-bacterial DNA pathogens, unless they mean that their dataset of DNA seq can be leveraged to look at the DNA viruses and fungi? Or that they already have a study coming out with these results? Please clarify this sentence.	We agree with the reviewer and have amended this statement: 'Future analysis will evaluate non-bacterial DNA-containing pathogens' has been replaced with 'Future analysis will evaluate these.'	Line 576
- Acknowledgments: no fundings are reported.	Thank you for highlighting this point. We have added these details to the revised manuscript, as well as a Competing Interests statement: 'This work was supported by the Australian National Health and Medical Research Council GNT1103262 (BAB), GNT1159151 (SP), GNT1158696 (SP) and GNT2009047 (SP), and by The Geok Hua Wong Charitable Trust. icddr,b is also grateful to the Governments of Bangladesh, Canada, Sweden, and the UK for providing core/unrestricted support. This work was made possible through Victorian State Government Operational Infrastructure Support and Australian Government NHMRC IRIISS. AB was supported by a Research Training Program Scholarship from the University of Melbourne and a stipend from WEHI.	Line 621

REVIEWER COMMENT	AUTHOR RESPONSE	MANUSCRIPT REFERENCE
	The study funder had no role in study design, data collection and analysis or manuscript writing. Competing interests statement The authors declare no competing interests.'	Line 673
- Data availability: when will data be public?	Sequencing data is now available and we have updated the manuscript to reflect this: 'The sequencing data used in this study have been deposited in the NCBI Sequence Read Archive (SRA) database under BioProject accession PRJNA1081952 [https://www.ncbi.nlm.nih.gov/sra]. Source data are provided with this paper.'	Line 653

Reviewer #3 (Remarks to the Author):

This is an interesting manuscript that leverages findings from a large RCT comparing iron supplementation with/without MNPs to placebo in a cohort of Bangladeshi children to evaluate changes in the microbiome associated with the intervention. The study adds some additional data to existing studies that have examined these associations, particularly in relation to concerns that iron may negatively impact the microbiome and predispose infants to pathogen growth and colonization.

REVIEWER COMMENT	AUTHOR RESPONSE	MANUSCRIPT REFERENCE
There are a number of limitations to this analysis that limit the overall impact of this study. Before discussing those specific issues, I believe there is a fundamental issue with authorship that must be clearly addressed. It appears that the trial was conducted in Bangladesh and a number of Bangladeshi authors contributed to the study design, implementation and write up. It is not clear why both the last and first authors are from high income institutions and why a Bangladeshi author is not acknowledged in a similar authorship position. Unless appropriate justification can be provided, it is important to ensure that colleagues in LMIC settings are given equal opportunity for first and last author positions – and given the apparent location and implementation needs of this study, this needs to be clearly explained.	We appreciate this point and the opportunity to provide more context regarding this manuscript and the BRISC trial. This program has been a partnership between Bangladeshi and Australian researchers working together for almost a decade. All authorship decisions were made on a shared basis between the study Principal Investigators: Drs Pasricha, Hamadani and Biggs. Note that for the main trial, Drs Biggs and Hamadani shared senior authorship. For the previous high impact publication from this trial, Dr Hamadani was first author and Dr Pasricha last author.¹ For the protocol paper, Dr Hasan was first author.² For the current study, we discussed authorship with the Bangladeshi authors and reached a decision on authorship order by consensus based on the scientific input for this particular project. Specifically, the first author Dr Baldi, an Australian PhD student, travelled to Bangladesh and worked with the field team to establish the collection and processing methods for the microbiome substudy. This included living in the field site for ~1 month	

REVIEWER COMMENT	AUTHOR RESPONSE	MANUSCRIPT REFERENCE
	and making regular visits. He also piloted all collection methods prior to setting up the main trial to confirm their suitability for this project. He organized material transfer agreements and government approval for shipping of samples, and organized sample shipping. In Australia, he personally extracted ~3000 stool samples, then undertook all 16S PCR and sequencing experiments. Likewise, he developed a method to downscale the QIAseq FX DNA Library Kit for shotgun metagenomics to 0.5x, enabling the large-scale study, then performed all library preparation and sequencing himself. He undertook all bioinformatic analysis. Ms Braat supported the statistical analysis of the trial overall, and the analysis of the microbiome data per arm as presented here. This work has extended ~4 years beyond the closing of the main trial. We recognize the contributions of the Bangladeshi team as co-authors on the paper; and appreciate their technical and logistical input into the study. This work was undertaken in the laboratory of the senior author, Dr Pasricha, who supervised and mentored the study design of the sample collection in the field and all wet analyses and bioinformatics in the lab. We have discussed the above with Drs Hamadani and Hasan, the key partners in Bangladesh, who agree with this approach. Note that Dr Hasan is a PhD student from icddr,b (Dhaka, Bangladesh) currently working in Dr Pasricha's lab, and will lead a number of first-author publications related to the project he is leading (related to child neurodevelopment).	

REVIEWER COMMENT	AUTHOR RESPONSE	MANUSCRIPT REFERENCE
The study has several key limitations; 1) This particular analysis did not focus on clinical outcomes – instead evaluated proxy measures of potential harm. As such, I do not think safety is appropriate to reference here. This is a study of potential adverse impact.	We thank the reviewer for this comment. We have removed the term 'safety' from the title of our manuscript and it now reads: To 'Effects of iron supplements and iron-containing micronutrient powders on the gut microbiome in Bangladeshi infants: a randomized controlled trial'.	Line 1
2) Not all diarrhea is due to infectious causes. Without assessment of functional attributes (permeability, barrier function, etc.) it is not possible to rule out adverse impacts on the gut from these interventions.	We have added the follow text to the Discussion section: 'Our study was not designed to measure any effects from iron that may cause changes to intestinal function unrelated to microbiota reprofiling (for example, impaired barrier function), which may also cause diarrhea.'	Line 577
4) It appears that selection of these children was based on the presence of follow up samples. This is hugely problematic and completely eliminates the randomization benefits of the study as well as introducing sampling bias. In fact, there appear to be meaningful differences in baseline characteristics between the arms (iron deficiency at baseline for example). This is likely the result of sampling bias.	We thank the reviewer for the opportunity to clarify our recruitment strategy. The main trial recruited over a period of 18 months (July 2017 to February 2019) and our sub study recruited over the final six months of this period. All children recruited to the main trial were offered inclusion in this sub study. Of the 1093 children all were recruited to the sub study. Of these, 923 (84%) provided a baseline stool sample. All of these participants were asked to provide stool samples at midline and endline. At midline, 810 (74%) of the total available children and 88% of those who provided a baseline sample provided a post-intervention sample. All of these samples underwent DNA extraction for 16S rRNA analysis. Remaining blinded, we then sought to select samples from children for whom samples at each of the three time points were available in order to perform shotgun metagenomic analysis on a canonical group of samples.	

REVIEWER COMMENT	AUTHOR RESPONSE	MANUSCRIPT REFERENCE
	We have compared baseline data of the main trial (see table below from main BRISC publication) to the baseline data of the sub study and note that the baseline data are similar including baseline differences between arms.	

Table 1. Household and Child Characteristics at Baseline.*

Characteristic	Iron Syrup (N=1101)	MNPs (N=1099)	Placebo (N=1100)
Household			
Median maternal education (IQR) — yr	8 (5–10)	8 (5–10)	8 (5–10)
Median paternal education (IQR) — yr	8 (5–10)	7 (5–9)	8 (5–10)
Household wealth index — no./total no. (%)			
Quintile 1: relative poorest	222/1101 (20.2)	224/1098 (20.4)	215/1099 (19.6)
Quintile 3: relative middle	213/1101 (19.3)	230/1098 (20.9)	211/1099 (19.2)
Quintile 5: relative wealthiest	222/1101 (20.2)	221/1098 (20.1)	212/1099 (19.3)
Household with food-secure status — no./total no. (%)†	884/1099 (80.4)	877/1093 (80.2)	866/1094 (79.2)
Child			
Demographic characteristic			
Female sex — no./total no. (%)	550/1101 (50.0)	548/1099 (49.9)	550/1100 (50.0)
Age — mo	8.0±0.3	8.0±0.3	8.0±0.3
Laboratory measure			
Hemoglobin level — g/dl	11.0±1.0	11.0±1.0	11.0±1.0
Anemia — no./total no. (%)‡	495/1072 (46.2)	461/1053 (43.8)	472/1063 (44.4)
Median ferritin level (IQR) — µg/liter	21.7 (11.7–38.5)	23.1 (13.1–38.5)	23.8 (12.8–39.3)
Iron deficiency — no./total no. (%)§	307/1033 (29.7)	273/1021 (26.7)	272/1026 (26.5)
Iron deficiency anemia — no./total no. (%)¶	225/1033 (21.8)	181/1021 (17.7)	188/1026 (18.3)
Median C-reactive protein level (IQR) — mg/liter	0.68 (0.32–2.02)	0.77 (0.32–2.49)	0.76 (0.34–2.18)
Inflammation — no./total no. (%)	112/1033 (10.8)	147/1021 (14.4)	128/1027 (12.5)
Child growth			
Length-for-age z score	-1.23±1.04	-1.20±1.00	-1.27±1.00
Weight-for-age z score	-0.58±1.04	-0.57±1.01	-0.63±1.01
Weight-for-length z score	0.24±1.03	0.22±1.02	0.21±1.01
Child development according to the Bayley-III scores**			
Cognitive composite score	96.2±7.8	96.0±7.8	96.6±7.7
Language composite score	88.3±7.2	88.1±7.1	88.8±7.0
Motor composite score	92.0±10.1	92.1±10.2	92.7±10.4

REVIEWER COMMENT	AUTHOR RESPONSE	MANUSCRIPT REFERENCE
	From Pasricha, S.R., et al. Benefits and Risks of Iron Interventions in Infants in Rural Bangladesh. N Engl J Med 385, 982-995 (2021).³ We have included baseline data of the shotgun metagenomic analysis sub-cohort in the Supplementary Material (Table S1) to allow the reader to appropriately review our results. We have made reference to this in the Results section of the manuscript.	Table S1 Line 323

REVIEWER COMMENT	AUTHOR RESPONSE	MANUSCRIPT REFERENCE																
5) Retention in the study was very poor which is problematic as it may have led to significant bias in the presented outcomes. Children who completed the study may be fundamentally different from those who did not. In addition, the benefits of randomization are attenuated when there is differential loss to follow up. Please present data comparing baseline characteristics among those who completed follow/did not complete follow up in each arm.	We considered this issue in the main trial and demonstrated that there was no imbalance with regard to trial arm and loss to follow up or withdrawal from study. In our microbiome analysis we have focused on participants at midline where, as stated above, there was a high retention of participants. We also note similar rates of retention between trial arms at the midline time point within the sub study cohort (see table below).     Iron MNPs Placebo     Baseline (900) 298 299 303   Midline (792) 254 (85% of baseline) 262 (88% of baseline) 276 (91% of baseline)   Endline (596) 183 198 195   		Iron	MNPs	Placebo	Baseline (900)	298	299	303	Midline (792)	254 (85% of baseline)	262 (88% of baseline)	276 (91% of baseline)	Endline (596)	183	198	195	
	Iron	MNPs	Placebo															
Baseline (900)	298	299	303															
Midline (792)	254 (85% of baseline)	262 (88% of baseline)	276 (91% of baseline)															
Endline (596)	183	198	195															

6) Adherence to the intervention was also problematic. First, adherence is not a baseline feature so should not be presented in Table 1. Second, the denominator to assess adherence is not the proportion of children taking the intervention who were retained in the study – it is the proportion of all enrolled who took the intervention. When applying this, adherence was quite low. As a result, it is very difficult to determine the true impact of the intervention in this study. In addition, the poor adherence is complicated by the timing of the sampling. If many children were not adherent, then you are comparing children who received the intervention sooner vs. later to the outcome assessment. While the authors did compare outcomes among children fully adherent and those not adherent, this is likely limited by power do detect such differences.	We have removed adherence from Table 1 as recommended and placed these data in the text of the Results section. We would like to take this opportunity to clarify how we defined adherence in this trial: X% of children randomized to a trial arm who took $\geq 70\%$ of doses. This includes iron syrup, MNPs and placebo in our blinded double-dummy study design (i.e. each children was required to take a syrup – iron or placebo – and an MNP – iron-containing or placebo – formulation each day). During the 13-week intervention period in the main trial parents were asked this question at each weekly visit: 'Did your child take both the study medicines in last 7 days?', and if the answer was negative the field worker went on to ask about adherence for each day of that week. The figure below indicates that per-week adherence among trial participants was consistent, and this is a strength of our study.	Table 1 Line 424
--	---	-----------------------------

REVIEWER COMMENT	AUTHOR RESPONSE	MANUSCRIPT REFERENCE																												
	Adherence to both syrup and sachet per week   <caption>Adherence to both syrup and sachet per week</caption>   Week Percentage    172.8% 272.1% 373.2% 472.4% 572.5% 672.2% 772.3% 872.9% 971.3% 1071.4% 1171.2% 1271.9% 1375.5%  	Week	Percentage	1	72.8%	2	72.1%	3	73.2%	4	72.4%	5	72.5%	6	72.2%	7	72.3%	8	72.9%	9	71.3%	10	71.4%	11	71.2%	12	71.9%	13	75.5%	
Week	Percentage																													
1	72.8%																													
2	72.1%																													
3	73.2%																													
4	72.4%																													
5	72.5%																													
6	72.2%																													
7	72.3%																													
8	72.9%																													
9	71.3%																													
10	71.4%																													
11	71.2%																													
12	71.9%																													
13	75.5%																													
7) The findings of reduced Bifidobacterium and Lactobacillus spp. are important given potential consequences for growth and sepsis. This should be discussed.																														

8) Another important potential concern is the propagation of AMR genes as a result of the intervention. This would be a valuable addition.

To evaluate this point, we have performed per-arm analyses of differential abundance of AMR genes (p-values adjusted for multiple comparisons). The two volcano plots below show no statistically significant differences in AMR gene abundance between iron and placebo (first figure) and MNPs and placebo (second figure)

Additionally, there were no differences in total ARM abundance (in RPKM) between trial arms at the midline time point:

REVIEWER COMMENT	AUTHOR RESPONSE	MANUSCRIPT REFERENCE																														
	Iron     Mean SD Min Max     AMR (RPKM) 3976.736 2926.104 927.6136 19072.27    MNPs     Mean SD Min Max     AMR (RPKM) 3876.548 3328.412 575.5026 39839.55    Placebo     Mean SD Min Max     AMR (RPKM) 4004.754 3416.836 875.6277 29969.44   		Mean	SD	Min	Max	AMR (RPKM)	3976.736	2926.104	927.6136	19072.27		Mean	SD	Min	Max	AMR (RPKM)	3876.548	3328.412	575.5026	39839.55		Mean	SD	Min	Max	AMR (RPKM)	4004.754	3416.836	875.6277	29969.44	
	Mean	SD	Min	Max																												
AMR (RPKM)	3976.736	2926.104	927.6136	19072.27																												
	Mean	SD	Min	Max																												
AMR (RPKM)	3876.548	3328.412	575.5026	39839.55																												
	Mean	SD	Min	Max																												
AMR (RPKM)	4004.754	3416.836	875.6277	29969.44																												

REVIEWER COMMENT	AUTHOR RESPONSE	MANUSCRIPT REFERENCE
9) The overall conclusions of the paper are not strong and could be revised.	We have updated the discussion and conclusions section in line with the feedback of Reviewer 1 and 3. Importantly, given the complexity of our findings, we are aiming to be cautious in the interpretation of our data and to prioritise safety. Our revised conclusion is as follows: “Ultimately, although our results do not definitively confirm that iron interventions adversely reprofile the infant gut microbiome, an adverse effect of iron on the gut microbiome remains biologically plausible. Thus, the risk-benefit of iron should be carefully considered before implementing these interventions.”	
Additional minor comments below; 1) Line 66 – it is not clear what the authors mean by “diarrheal disease risk”.	We have simplified this to read 'diarrhea' rather than 'diarrheal disease risk': 'We identified no increase in diarrhea with either treatment.'	Line 75
2) Line 382 – the authors state that these changes are not due to false discovery. This cannot be determined definitively and this statement should be modified – it is overstated.	Thank you for this comment. We have amended the final sentence in the relevant paragraph to read: 'These findings are supported by evidence from shotgun metagenomic data of differential enrichment of two phyla previously linked to iron-microbiome interactions, suggesting these changes may not be due to false discovery errors.'	Line 454

References

1. Hamadani, J.D., et al. Immediate impact of stay-at-home orders to control COVID-19 transmission on socioeconomic conditions, food insecurity, mental health, and intimate partner violence in Bangladeshi women and their families: an interrupted time series. *Lancet Glob Health* 8, e1380-e1389 (2020).
2. Hasan, M.I., et al. Benefits and risks of Iron interventions in children (BRISC): protocol for a three-arm parallel-group randomised controlled field trial in Bangladesh. *BMJ Open* 7, e018325 (2017).
3. Pasricha, S.-R., et al. Benefits and Risks of Iron Interventions in Infants in Rural Bangladesh. *New England Journal of Medicine* 385, 982-995 (2021).

REVIEWERS' COMMENTS

Reviewer #3 (Remarks to the Author):

The authors have done a nice job responding to reviewer comments and clarifying many of the issues raised. The issue of authorship was clarified appropriately and it is clear that the collaborative partners have taken the time to discuss and agree on authorship criteria. Thank you for detailing those discussions.

The study is limited by potential bias introduced by the inclusion of only those with complete samples and the suboptimal adherence. The authors address these issues but they remain limitations to the interpretation of these data.